# Efficient Neural Network Robustness Certification with General Activation Functions

**Huan Zhang**[1,†,*]   **Tsui-Wei Weng**[2,†]   **Pin-Yu Chen**[3]   **Cho-Jui Hsieh**[1]   **Luca Daniel**[2]

[1]University of California, Los Angeles, Los Angeles CA 90095
[2]Massachusetts Institute of Technology, Cambridge, MA 02139
[3]MIT-IBM Watson AI Lab, IBM Research, Yorktown Heights, NY 10598
`huan@huan-zhang.com, twweng@mit.edu`
`pin-yu.chen@ibm.com, chohsieh@cs.ucla.edu, dluca@mit.edu`

## Abstract

Finding minimum distortion of adversarial examples and thus certifying robustness in neural network classifiers for given data points is known to be a challenging problem. Nevertheless, recently it has been shown to be possible to give a non-trivial certified lower bound of minimum adversarial distortion, and some recent progress has been made towards this direction by exploiting the piece-wise linear nature of ReLU activations. However, a generic robustness certification for *general* activation functions still remains largely unexplored. To address this issue, in this paper we introduce CROWN, a general framework to certify robustness of neural networks with general activation functions for given input data points. The novelty in our algorithm consists of bounding a given activation function with linear and quadratic functions, hence allowing it to tackle general activation functions including but not limited to four popular choices: ReLU, tanh, sigmoid and arctan. In addition, we facilitate the search for a tighter certified lower bound by *adaptively* selecting appropriate surrogates for each neuron activation. Experimental results show that CROWN on ReLU networks can notably improve the certified lower bounds compared to the current state-of-the-art algorithm Fast-Lin, while having comparable computational efficiency. Furthermore, CROWN also demonstrates its effectiveness and flexibility on networks with general activation functions, including tanh, sigmoid and arctan.

## 1   Introduction

While neural networks (NNs) have achieved remarkable performance and accomplished unprecedented breakthroughs in many machine learning tasks, recent studies have highlighted their lack of robustness against adversarial perturbations [1, 2]. For example, in image learning tasks such as object classification [3, 4, 5, 6] or content captioning [7], visually indistinguishable adversarial examples can be easily crafted from natural images to alter a NN's prediction result. Beyond the white-box attack setting where the target model is entirely transparent, visually imperceptible adversarial perturbations can also be generated in the black-box setting by only using the prediction results of the target model [8, 9, 10, 11]. In addition, real-life adversarial examples have been made possible through the lens of realizing physical perturbations [12, 13, 14]. As NNs are becoming a core technique deployed in a wide range of applications, including safety-critical tasks, certifying robustness of a NN against adversarial perturbations has become an important research topic in machine learning.

Correspondence to Huan Zhang <`huan@huan-zhang.com`> and Tsui-Wei Weng <`twweng@mit.edu`>

Given a NN (possibly with a deep and complicated network architecture), we are interested in certifying the (local) robustness of an arbitrary natural example $\mathbf{x}_0$ by ensuring *all* its neighborhood has the same inference outcome (e.g., consistent top-1 prediction). In this paper, the neighborhood of $\mathbf{x}_0$ is characterized by an $\ell_p$ ball centered at $\mathbf{x}_0$, for any $p \geq 1$. Geometrically speaking, the minimum distance of a misclassified nearby example to $\mathbf{x}_0$ is the least adversary strength (a.k.a. minimum adversarial distortion) required to alter the target model's prediction, which is also the largest possible robustness certificate for $\mathbf{x}_0$. Unfortunately, finding the minimum distortion of adversarial examples in NNs with Rectified Linear Unit (ReLU) activations, which is one of the most widely used activation functions, is known to be an NP-complete problem [15, 16]. This makes formal verification techniques such as Reluplex [15] computationally demanding even for small-sized NNs and suffer from scalability issues.

Although certifying the largest possible robustness is challenging for ReLU networks, the piecewise linear nature of ReLUs can be exploited to efficiently compute a non-trivial *certified lower bound* of the minimum distortion [17, 18, 19, 20]. Beyond ReLU, one fundamental problem that remains largely unexplored is how to generalize the robustness certification technique to other popular activation functions that are not piece-wise linear, such as tanh and sigmoid, and how to motivate and certify the design of other activation functions towards improved robustness. In this paper, we tackle the preceding problem by proposing an efficient robustness certification framework for NNs with general activation functions. Our main contributions in this paper are summarized as follows:

- We propose a generic analysis framework CROWN for certifying NNs using linear or quadratic upper and lower bounds for *general* activation functions that are not necessarily piece-wise linear.
- Unlike previous work [20], CROWN allows flexible selections of upper/lower bounds for activation functions, enabling an *adaptive* scheme that helps to reduce approximation errors. Our experiments show that CROWN achieves up to 26% improvements in certified lower bounds compared to [20].
- Our algorithm is efficient and can scale to large NNs with various activation functions. For a NN with over 10,000 neurons, we can give a certified lower bound in about 1 minute on 1 CPU core.

## 2   Background and Related Work

For ReLU networks, finding the minimum adversarial distortion for a given input data point $x_0$ can be cast as a mixed integer linear programming (MILP) problem [21, 22, 23]. Reluplex [15, 24] uses a satisfiable modulo theory (SMT) to encode ReLU activations into linear constraints. Similarly, Planet [25] uses satisfiability (SAT) solvers. However, due to the NP-completeness for solving such a problem [15], these methods can only find minimum distortion for very small networks. It can take Reluplex several hours to find the minimum distortion of an example for a ReLU network with 5 inputs, 5 outputs and 300 neurons[15].

On the other hand, a computationally feasible alternative of robustness certificate is to provide a non-trivial and *certified lower bound* of minimum distortion. Some analytical lower bounds based on operator norms on the weight matrices [3] or the Jacobian matrix in NNs [17] do not exploit special property of ReLU and thus can be very loose [20]. The bounds in [26, 27] are based on the local Lipschitz constant. [26] assumes a continuous differentiable NN and hence excludes ReLU networks; a closed form lower-bound is also hard to derive for networks beyond 2 layers. [27] applies to ReLU networks and uses Extreme Value Theory to provide an estimated lower bound (CLEVER score). Although the CLEVER score is capable of reflecting the level of robustness in different NNs and is scalable to large networks, it is not a certified lower bound. On the other hand, [18] use the idea of a convex outer adversarial polytope in ReLU networks to compute a certified lower bound by relaxing the MILP certification problem to linear programing (LP). [20] exploit the ReLU property to bound the activation function (or the local Lipschitz constant) and provide efficient algorithms (Fast-Lin and Fast-Lip) for computing a certified lower bound, achieving state-of-the-art performance. A recent work [28] uses abstract transformations to zonotopes for proving robustness property for ReLU networks. Nonetheless, there are still some applications demand non-ReLU activations, e.g. RNN and LSTM, thus a general framework that can efficiently compute non-trivial and certified lower bounds for NNs with general activation functions is of great importance. We aim at filling this gap and propose CROWN that can perform efficient robustness certification to NNs with general activation functions. Table 1 summarizes the differences of other approaches and CROWN. Note that the semidefinite programming approach proposed in [19] and a recent work [29] based on solving Lagrangian dual can both handle general activation functions, but [19] is limited to NNs with one hidden layer and [29] trades off the quality of robustness bound with scalability.

Table 1: Comparison of methods for providing adversarial robustness certification in NNs.

| Method | Non-trivial bound | Multi-layer | Scalability | Beyond ReLU |
|---|---|---|---|---|
| Szegedy et. al. [3] | × | ✓ | ✓ | ✓ |
| Reluplex [15], Planet [25] | ✓ | ✓ | × | × |
| Hein & Andriushchenko [26] | ✓ | × | ✓ | differentiable[*] |
| Raghunathan et al. [19] | ✓ | × | × | ✓ |
| Kolter and Wong [18] | ✓ | ✓ | ✓ | × |
| Fast-lin / Fast-lip [20] | ✓ | ✓ | ✓ | × |
| CROWN (ours) | ✓ | ✓ | ✓ | ✓ (general) |

[*] Continuously differentiable activation function required (soft-plus is demonstrated in [26])

Some recent works (such as robust optimization based adversarial training [30] or region-based classification [31]) *empirically* exhibit strong robustness against several adversarial attacks, which is beyond the scope of *provable* robustness certification. In addition, Sinha et al. [16] provide distributional robustness certification based on Wasserstein distance between data distributions, which is different from the local $\ell_p$ ball robustness model considered in this paper.

## 3    CROWN: A general framework for certifying neural networks

**Overview of our results.**    In this section, we present a general framework CROWN for efficiently computing a certified lower bound of minimum adversarial distortion given any input data point $x_0$ with general activation functions in larger NNs. We first provide principles in Section 3.1 to derive output bounds of NNs when the inputs are perturbed within an $\ell_p$ ball and each neuron has different (adaptive) linear approximation bounds on its activation function. In Section 3.2, we demonstrate how to provide robustness certification for four widely-used activation functions (ReLU, tanh, sigmoid and arctan) using CROWN. In particular, we show that the state-of-the-art Fast-Lin algorithm is a special case under the CROWN framework and that the adaptive selections of approximation bounds allow CROWN to achieve a tighter (larger) certified lower bound (see Section 4). In Section 3.3, we further highlight the flexibility of CROWN to incorporate quadratic approximations on the activation functions in addition to the linear approximations described in Section 3.1.

### 3.1    General framework

**Notations.**    For an $m$-layer neural network with an input vector $\mathbf{x} \in \mathbb{R}^{n_0}$, let the number of neurons in each layer be $n_k, \forall k \in [m]$, where $[i]$ denotes set $\{1, 2, \cdots, i\}$. Let the $k$-th layer weight matrix be $\mathbf{W}^{(k)} \in \mathbb{R}^{n_k \times n_{k-1}}$ and bias vector be $\mathbf{b}^{(k)} \in \mathbb{R}^{n_k}$, and let $\Phi_k : \mathbb{R}^{n_0} \rightarrow \mathbb{R}^{n_k}$ be the operator mapping from input to layer $k$. We have $\Phi_k(\mathbf{x}) = \sigma(\mathbf{W}^{(k)}\Phi_{k-1}(\mathbf{x}) + \mathbf{b}^{(k)}), \forall k \in [m-1]$, where $\sigma(\cdot)$ is the coordinate-wise activation function. While our methodology is applicable to any activation function of interest, we emphasize on four most widely-used activation functions, namely ReLU: $\sigma(y) = \max(y, 0)$, hyperbolic tangent: $\sigma(y) = \tanh(y)$, sigmoid: $\sigma(y) = 1/(1 + e^{-y})$ and arctan: $\sigma(y) = \arctan(y)$. Note that the input $\Phi_0(\mathbf{x}) = \mathbf{x}$, and the vector output of the NN is $f(\mathbf{x}) = \Phi_m(\mathbf{x}) = \mathbf{W}^{(m)}\Phi_{m-1}(\mathbf{x}) + \mathbf{b}^{(m)}$. The $j$-th output element is denoted as $f_j(\mathbf{x}) = [\Phi_m(\mathbf{x})]_j$.

**Input perturbation and pre-activation bounds.**    Let $\mathbf{x_0} \in \mathbb{R}^{n_0}$ be a given data point, and let the perturbed input vector $\mathbf{x}$ be within an $\epsilon$-bounded $\ell_p$-ball centered at $\mathbf{x_0}$, i.e., $\mathbf{x} \in \mathbb{B}_p(\mathbf{x_0}, \epsilon)$, where $\mathbb{B}_p(\mathbf{x_0}, \epsilon) := \{\mathbf{x} \mid \|\mathbf{x} - \mathbf{x_0}\|_p \leq \epsilon\}$. For the $r$-th neuron in $k$-th layer, let its *pre-activation* input be $\mathbf{y}_r^{(k)}$, where $\mathbf{y}_r^{(k)} = \mathbf{W}_{r,:}^{(k)}\Phi_{k-1}(\mathbf{x}) + \mathbf{b}_r^{(k)}$ and $\mathbf{W}_{r,:}^{(k)}$ denotes the $r$-th row of matrix $\mathbf{W}^{(k)}$. When $\mathbf{x_0}$ is perturbed within an $\epsilon$-bounded $\ell_p$-ball, let $\mathbf{l}_r^{(k)}, \mathbf{u}_r^{(k)} \in \mathbb{R}$ be the pre-activation lower bound and upper bound of $\mathbf{y}_r^{(k)}$, i.e. $\mathbf{l}_r^{(k)} \leq \mathbf{y}_r^{(k)} \leq \mathbf{u}_r^{(k)}$.

Below, we first define the linear upper bounds and lower bounds of activation functions in Definition 3.1, which are the key to derive explicit output bounds for an $m$-layer neural network with general activation functions. The formal statement of the explicit output bounds is shown in Theorem 3.2.

**Definition 3.1** (Linear bounds on activation function). *For the $r$-th neuron in $k$-th layer with pre-activation bounds $\mathbf{l}_r^{(k)}, \mathbf{u}_r^{(k)}$ and the activation function $\sigma(y)$, define two linear functions $h_{U,r}^{(k)}, h_{L,r}^{(k)} : \mathbb{R} \rightarrow \mathbb{R}, h_{U,r}^{(k)}(y) = \alpha_{U,r}^{(k)}(y + \beta_{U,r}^{(k)}), h_{L,r}^{(k)}(y) = \alpha_{L,r}^{(k)}(y + \beta_{L,r}^{(k)}), such that $h_{L,r}^{(k)}(y) \leq \sigma(y) \leq h_{U,r}^{(k)}(y), y \in [\mathbf{l}_r^{(k)}, \mathbf{u}_r^{(k)}], \forall k \in [m-1], r \in [n_k]$ and $\alpha_{U,r}^{(k)}, \alpha_{L,r}^{(k)} \in \mathbb{R}^+, \beta_{U,r}^{(k)}, \beta_{L,r}^{(k)} \in \mathbb{R}$.*

Note that the parameters $\alpha_{U,r}^{(k)}, \alpha_{L,r}^{(k)}, \beta_{U,r}^{(k)}, \beta_{L,r}^{(k)}$ depend on $\mathbf{l}_r^{(k)}$ and $\mathbf{u}_r^{(k)}$, i.e. for different $\mathbf{l}_r^{(k)}$ and $\mathbf{u}_r^{(k)}$ we may choose different parameters. Also, for ease of exposition, in this paper we restrict $\alpha_{U,r}^{(k)}, \alpha_{L,r}^{(k)} \geq 0$. However, Theorem 3.2 can be easily generalized to the case of negative $\alpha_{U,r}^{(k)}, \alpha_{L,r}^{(k)}$.

**Theorem 3.2** (Explicit output bounds of neural network $f$). *Given an $m$-layer neural network function $f : \mathbb{R}^{n_0} \to \mathbb{R}^{n_m}$, there exists two explicit functions $f_j^L : \mathbb{R}^{n_0} \to \mathbb{R}$ and $f_j^U : \mathbb{R}^{n_0} \to \mathbb{R}$ such that $\forall j \in [n_m]$, $\forall \mathbf{x} \in \mathbb{B}_p(\mathbf{x_0}, \epsilon)$, the inequality $f_j^L(\mathbf{x}) \leq f_j(\mathbf{x}) \leq f_j^U(\mathbf{x})$ holds true, where*

$$f_j^U(\mathbf{x}) = \mathbf{\Lambda}_{j,:}^{(0)} \mathbf{x} + \sum_{k=1}^{m} \mathbf{\Lambda}_{j,:}^{(k)}(\mathbf{b}^{(k)} + \mathbf{\Delta}_{:,j}^{(k)}), \quad f_j^L(\mathbf{x}) = \mathbf{\Omega}_{j,:}^{(0)} \mathbf{x} + \sum_{k=1}^{m} \mathbf{\Omega}_{j,:}^{(k)}(\mathbf{b}^{(k)} + \mathbf{\Theta}_{:,j}^{(k)}), \quad (1)$$

$$\mathbf{\Lambda}_{j,:}^{(k-1)} = \begin{cases} \mathbf{e}_j^\top & \text{if } k = m+1; \\ (\mathbf{\Lambda}_{j,:}^{(k)} \mathbf{W}^{(k)}) \odot \lambda_{j,:}^{(k-1)} & \text{if } k \in [m]. \end{cases} \quad \mathbf{\Omega}_{j,:}^{(k-1)} = \begin{cases} \mathbf{e}_j^\top & \text{if } k = m+1; \\ (\mathbf{\Omega}_{j,:}^{(k)} \mathbf{W}^{(k)}) \odot \omega_{j,:}^{(k-1)} & \text{if } k \in [m]. \end{cases}$$

*and $\forall i \in [n_k]$, we define four matrices $\lambda^{(k)}, \omega^{(k)}, \mathbf{\Delta}^{(k)}, \mathbf{\Theta}^{(k)} \in \mathbb{R}^{n_m \times n_k}$:*

$$\lambda_{j,i}^{(k)} = \begin{cases} \alpha_{U,i}^{(k)} & \text{if } k \neq 0, \mathbf{\Lambda}_{j,:}^{(k+1)} \mathbf{W}_{:,i}^{(k+1)} \geq 0; \\ \alpha_{L,i}^{(k)} & \text{if } k \neq 0, \mathbf{\Lambda}_{j,:}^{(k+1)} \mathbf{W}_{:,i}^{(k+1)} < 0; \\ 1 & \text{if } k = 0. \end{cases} \qquad \omega_{j,i}^{(k)} = \begin{cases} \alpha_{L,i}^{(k)} & \text{if } k \neq 0, \mathbf{\Omega}_{j,:}^{(k+1)} \mathbf{W}_{:,i}^{(k+1)} \geq 0; \\ \alpha_{U,i}^{(k)} & \text{if } k \neq 0, \mathbf{\Omega}_{j,:}^{(k+1)} \mathbf{W}_{:,i}^{(k+1)} < 0; \\ 1 & \text{if } k = 0. \end{cases}$$

$$\mathbf{\Delta}_{i,j}^{(k)} = \begin{cases} \beta_{U,i}^{(k)} & \text{if } k \neq m, \mathbf{\Lambda}_{j,:}^{(k+1)} \mathbf{W}_{:,i}^{(k+1)} \geq 0; \\ \beta_{L,i}^{(k)} & \text{if } k \neq m, \mathbf{\Lambda}_{j,:}^{(k+1)} \mathbf{W}_{:,i}^{(k+1)} < 0; \\ 0 & \text{if } k = m. \end{cases} \qquad \mathbf{\Theta}_{i,j}^{(k)} = \begin{cases} \beta_{L,i}^{(k)} & \text{if } k \neq m, \mathbf{\Omega}_{j,:}^{(k+1)} \mathbf{W}_{:,i}^{(k+1)} \geq 0; \\ \beta_{U,i}^{(k)} & \text{if } k \neq m, \mathbf{\Omega}_{j,:}^{(k+1)} \mathbf{W}_{:,i}^{(k+1)} < 0; \\ 0 & \text{if } k = m. \end{cases}$$

*and $\odot$ is the Hadamard product and $\mathbf{e}_j \in \mathbb{R}^{n_m}$ is a standard unit vector at $j$th coordinate .*

Theorem 3.2 illustrates how a NN function $f_j(\mathbf{x})$ can be bounded by two linear functions $f_j^U(\mathbf{x})$ and $f_j^L(\mathbf{x})$ when the activation function of each neuron is bounded by two linear functions $h_{U,r}^{(k)}$ and $h_{L,r}^{(k)}$ in Definition 3.1. The central idea is to unwrap the activation functions layer by layer by considering the signs of the associated (equivalent) weights of each neuron and apply the two linear bounds $h_{U,r}^{(k)}$ and $h_{L,r}^{(k)}$. As we demonstrate in the proof, when we replace the activation functions with the corresponding linear upper bounds and lower bounds at the layer $m - 1$, we can then define equivalent weights and biases based on the parameters of $h_{U,r}^{(m-1)}$ and $h_{L,r}^{(m-1)}$ (e.g. $\mathbf{\Lambda}^{(k)}, \mathbf{\Delta}^{(k)}, \mathbf{\Omega}^{(k)}, \mathbf{\Theta}^{(k)}$ are related to the terms $\alpha_{U,r}^{(k)}, \beta_{U,r}^{(k)}, \alpha_{L,r}^{(k)}, \beta_{L,r}^{(k)}$, respectively) and then repeat the procedure to "back-propagate" to the input layer. This allows us to obtain $f_j^U(\mathbf{x})$ and $f_j^L(\mathbf{x})$ in (1). The formal proof of Theorem 3.2 is in Appendix A. Note that for a neuron $r$ in layer $k$, the slopes of its linear upper and lower bounds $\alpha_{U,r}^{(k)}, \alpha_{L,r}^{(k)}$ can be different. This implies:

1. Fast-Lin [20] is a special case of our framework as they require the slopes $\alpha_{U,r}^{(k)}, \alpha_{L,r}^{(k)}$ to be the same; and it only applies to ReLU networks (cf. Sec. 3.2). In Fast-Lin, $\mathbf{\Lambda}^{(0)}$ and $\mathbf{\Omega}^{(0)}$ are identical.

2. Our CROWN framework allows *adaptive selections* on the linear approximation when computing certified lower bounds of minimum adversarial distortion, which is the main contributor to improve the certified lower bound as demonstrated in the experiments in Section 4.

**Global bounds.** More importantly, since the input $\mathbf{x} \in \mathbb{B}_p(\mathbf{x_0}, \epsilon)$, we can take the maximum, i.e. $\max_{\mathbf{x} \in \mathbb{B}_p(\mathbf{x_0}, \epsilon)} f_j^U(\mathbf{x})$, and minimum, i.e. $\min_{\mathbf{x} \in \mathbb{B}_p(\mathbf{x_0}, \epsilon)} f_j^L(\mathbf{x})$, as a pair of global upper and lower bound of $f_j(\mathbf{x})$ – which in fact has *closed-form* solutions because $f_j^U(\mathbf{x})$ and $f_j^L(\mathbf{x})$ are two linear functions and $\mathbf{x} \in \mathbb{B}_p(\mathbf{x_0}, \epsilon)$ is a convex norm constraint. This result is formally presented below and its proof is given in Appendix B.

**Corollary 3.3** (Closed-form global bounds). *Given a data point $\mathbf{x_0} \in \mathbb{R}^{n_0}$, $\ell_p$ ball parameters $p \geq 1$ and $\epsilon > 0$. For an $m$-layer neural network function $f : \mathbb{R}^{n_0} \to \mathbb{R}^{n_m}$, there exists two fixed values $\gamma_j^L$ and $\gamma_j^U$ such that $\forall \mathbf{x} \in \mathbb{B}_p(\mathbf{x_0}, \epsilon)$ and $\forall j \in [n_m]$, $1/q = 1 - 1/p$, the inequality $\gamma_j^L \leq f_j(\mathbf{x}) \leq \gamma_j^U$ holds true, where*

$$\gamma_j^U = \epsilon \|\mathbf{\Lambda}_{j,:}^{(0)}\|_q + \mathbf{\Lambda}_{j,:}^{(0)} \mathbf{x_0} + \sum_{k=1}^{m} \mathbf{\Lambda}_{j,:}^{(k)}(\mathbf{b}^{(k)} + \mathbf{\Delta}_{:,j}^{(k)}), \ \gamma_j^L = -\epsilon \|\mathbf{\Omega}_{j,:}^{(0)}\|_q + \mathbf{\Omega}_{j,:}^{(0)} \mathbf{x_0} + \sum_{k=1}^{m} \mathbf{\Omega}_{j,:}^{(k)}(\mathbf{b}^{(k)} + \mathbf{\Theta}_{:,j}^{(k)}).$$

$$(2)$$

Table 2: Linear upper bound parameters of various activation functions: $h_{U,r}^{(k)}(y) = \alpha_{U,r}^{(k)}(y + \beta_{U,r}^{(k)})$

| **Upper bound** $h_{U,r}^{(k)}$ for activation function | $r \in \mathcal{S}_k^+$ | | $r \in \mathcal{S}_k^-$ | | $r \in \mathcal{S}_k^\pm$ | |
|---|---|---|---|---|---|---|
| | $\alpha_{U,r}^{(k)}$ | $\beta_{U,r}^{(k)}$ | $\alpha_{U,r}^{(k)}$ | $\beta_{U,r}^{(k)}$ | $\alpha_{U,r}^{(k)}$ | $\beta_{U,r}^{(k)}$ |
| ReLU | $1$ | $0$ | $0$ | $0$ | $a$ $(a \geq \frac{\mathbf{u}_r^{(k)}}{\mathbf{u}_r^{(k)} - \mathbf{l}_r^{(k)}}, \text{e.g. } a = \frac{\mathbf{u}_r^{(k)}}{\mathbf{u}_r^{(k)} - \mathbf{l}_r^{(k)}})$ | $-\mathbf{l}_r^{(k)}$ |
| Sigmoid, tanh (denoted as $\sigma(y)$) | $\sigma'(d)$ $(\mathbf{l}_r^{(k)} \leq d \leq \mathbf{u}_r^{(k)})$ | $\frac{\sigma(d)}{\alpha_{U,r}^{(k)}} - d$ * | $\frac{\sigma(\mathbf{u}_r^{(k)}) - \sigma(\mathbf{l}_r^{(k)})}{\mathbf{u}_r^{(k)} - \mathbf{l}_r^{(k)}}$ | $\frac{\sigma(\mathbf{l}_r^{(k)})}{\alpha_{U,r}^{(k)}} - \mathbf{l}_r^{(k)}$ | $\sigma'(d)$ $(\frac{\sigma(d) - \sigma(\mathbf{l}_r^{(k)})}{d - \mathbf{l}_r^{(k)}} - \sigma'(d) = 0, d \geq 0)$ ◇ | $\frac{\sigma(\mathbf{l}_r^{(k)})}{\alpha_{U,r}^{(k)}} - \mathbf{l}_r^{(k)}$ |

\* If $\alpha_{U,r}^{(k)}$ is close to 0, we suggest to calculate the intercept directly, $\alpha_{U,r}^{(k)} \cdot \beta_{U,r}^{(k)} = \sigma(d) - \alpha_{U,r}^{(k)}d$, to avoid numerical issues in implementation. Same for other similar cases.

◇ Alternatively, if the solution $d \geq \mathbf{u}_r^{(k)}$, then we can set $\alpha_{U,r}^{(k)} = \frac{\sigma(\mathbf{u}_r^{(k)}) - \sigma(\mathbf{l}_r^{(k)})}{\mathbf{u}_r^{(k)} - \mathbf{l}_r^{(k)}}$.

Table 3: Linear lower bound parameters of various activation functions: $h_{L,r}^{(k)}(y) = \alpha_{L,r}^{(k)}(y + \beta_{L,r}^{(k)})$

| **Lower bound** $h_{L,r}^{(k)}$ for activation function | $r \in \mathcal{S}_k^+$ | | $r \in \mathcal{S}_k^-$ | | $r \in \mathcal{S}_k^\pm$ | |
|---|---|---|---|---|---|---|
| | $\alpha_{L,r}^{(k)}$ | $\beta_{L,r}^{(k)}$ | $\alpha_{L,r}^{(k)}$ | $\beta_{L,r}^{(k)}$ | $\alpha_{L,r}^{(k)}$ | $\beta_{L,r}^{(k)}$ |
| ReLU | $1$ | $0$ | $0$ | $0$ | $a$ $(0 \leq a \leq 1, \text{e.g. } a = \frac{\mathbf{u}_r^{(k)}}{\mathbf{u}_r^{(k)} - \mathbf{l}_r^{(k)}}, 0, 1)$ | $0$ |
| Sigmoid, tanh (denoted as $\sigma(y)$) | $\frac{\sigma(\mathbf{u}_r^{(k)}) - \sigma(\mathbf{l}_r^{(k)})}{\mathbf{u}_r^{(k)} - \mathbf{l}_r^{(k)}}$ | $\frac{\sigma(\mathbf{l}_r^{(k)})}{\alpha_{L,r}^{(k)}} - \mathbf{l}_r^{(k)}$ | $\sigma'(d)$ $(\mathbf{l}_r^{(k)} \leq d \leq \mathbf{u}_r^{(k)})$ | $\frac{\sigma(d)}{\alpha_{L,r}^{(k)}} - d$ | $\sigma'(d)$ $(\frac{\sigma(d) - \sigma(\mathbf{u}_r^{(k)})}{d - \mathbf{u}_r^{(k)}} - \sigma'(d) = 0, d \leq 0)$ † | $\frac{\sigma(\mathbf{u}_r^{(k)})}{\alpha_{L,r}^{(k)}} - \mathbf{u}_r^{(k)}$ |

† Alternatively, if the solution $d \leq \mathbf{l}_r^{(k)}$, then we can set $\alpha_{L,r}^{(k)} = \frac{\sigma(\mathbf{u}_r^{(k)}) - \sigma(\mathbf{l}_r^{(k)})}{\mathbf{u}_r^{(k)} - \mathbf{l}_r^{(k)}}$.

**Certified lower bound of minimum distortion.** Given an input example $\mathbf{x_0}$ and an $m$-layer NN, let $c$ be the predicted class of $\mathbf{x_0}$ and $t \neq c$ be the targeted attack class. We aim to use the uniform bounds established in Corollary 3.3 to obtain the largest possible lower bound $\tilde{\epsilon}_t$ and $\tilde{\epsilon}$ of targeted and untargeted attacks respectively, which can be formulated as follows:

$$\tilde{\epsilon}_t = \max_\epsilon \epsilon \text{ s.t. } \gamma_c^L(\epsilon) - \gamma_t^U(\epsilon) > 0 \text{ and } \tilde{\epsilon} = \min_{t \neq c} \tilde{\epsilon}_t.$$

We note that although there is a linear $\epsilon$ term in (2), other terms such as $\mathbf{\Lambda}^{(k)}, \mathbf{\Delta}^{(k)}$ and $\mathbf{\Omega}^{(k)}, \mathbf{\Theta}^{(k)}$ also implicitly depend on $\epsilon$. This is because the parameters $\alpha_{U,i}^{(k)}, \beta_{U,i}^{(k)}, \alpha_{L,i}^{(k)}, \beta_{L,i}^{(k)}$ depend on $\mathbf{l}_i^{(k)}, \mathbf{u}_i^{(k)}$, which may vary with $\epsilon$; thus the values in $\mathbf{\Lambda}^{(k)}, \mathbf{\Delta}^{(k)}, \mathbf{\Omega}^{(k)}, \mathbf{\Theta}^{(k)}$ depend on $\epsilon$. It is therefore difficult to obtain an explicit expression of $\gamma_c^L(\epsilon) - \gamma_t^U(\epsilon)$ in terms of $\epsilon$. Fortunately, we can still perform a binary search to obtain $\tilde{\epsilon}_t$ with Corollary 3.3. More precisely, we first initialize $\epsilon$ at some fixed positive value and apply Corollary 3.3 repeatedly to obtain $\mathbf{l}_r^{(k)}$ and $\mathbf{u}_r^{(k)}$ from $k = 1$ to $m$ and $r \in [n_k]$. We then check if the condition $\gamma_c^L - \gamma_t^U > 0$ is satisfied. If so, we increase $\epsilon$; otherwise, we decrease $\epsilon$; and we repeat the procedure until a given tolerance level is met.[2]

**Time Complexity.** With Corollary 3.3, we can compute analytic output bounds efficiently without resorting to any optimization solvers for general $\ell_p$ distortion, and the time complexity for an $m$-layer ReLU network is polynomial time in contrast to Reluplex or Mixed-Integer Optimization-based approach [22, 23] where SMT and MIO solvers are exponential-time. For an $m$ layer network with $n$ neurons per layer and $n$ outputs, time complexity of CROWN is $O(m^2n^3)$. Forming $\mathbf{\Lambda}^{(0)}$ and $\mathbf{\Omega}^{(0)}$ for the $m$-th layer involves multiplications of layer weights in a similar cost of forward propagation in $O(mn^3)$ time. Also, the bounds for all previous $k \in [m-1]$ layers need to be computed beforehand in $O(kn^3)$ time; thus the total time complexity is $O(m^2n^3)$.

### 3.2 Case studies: CROWN for ReLU, tanh, sigmoid and arctan activations

In Section 3.1 we showed that as long as one can identify two linear functions $h_U(y), h_L(y)$ to bound a general activation function $\sigma(y)$ for each neuron, we can use Corollary 3.3 with a binary search

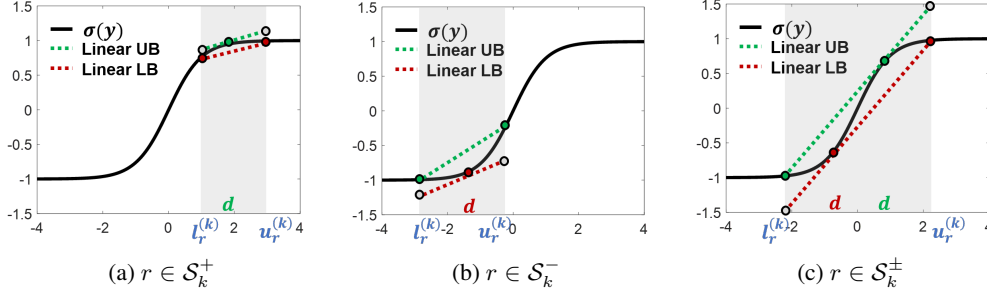

|  | | |
|---|---|---|
| (a) $r \in \mathcal{S}_k^+$ | (b) $r \in \mathcal{S}_k^-$ | (c) $r \in \mathcal{S}_k^{\pm}$ |

Figure 1: $\sigma(y) = \tanh$. Green lines are the upper bounds $h_{U,r}^{(k)}$; red lines are the lower bounds $h_{L,r}^{(k)}$.

to obtain certified lower bounds of minimum distortion. In this section, we illustrate how to find parameters $\alpha_{U,r}^{(k)}, \alpha_{L,r}^{(k)}$ and $\beta_{U,r}^{(k)}, \beta_{L,r}^{(k)}$ of $h_U(y), h_L(y)$ for four most widely used activation functions: ReLU, tanh, sigmoid and arctan. Other activations, including but not limited to leaky ReLU, ELU and softplus, can be easily incorporated into our CROWN framework following a similar procedure.

**Segmenting activation functions.** Based on the signs of $\mathbf{l}_r^{(k)}$ and $\mathbf{u}_r^{(k)}$, we define a partition $\{\mathcal{S}_k^+, \mathcal{S}_k^{\pm}, \mathcal{S}_k^-\}$ of set $[n_k]$ such that every neuron in $k$-th layer belongs to exactly one of the three sets. The formal definition of $\mathcal{S}_k^+, \mathcal{S}_k^{\pm}$ and $\mathcal{S}_k^-$ is $\mathcal{S}_k^+ = \{r \in [n_k] \mid 0 \leq \mathbf{l}_r^{(k)} \leq \mathbf{u}_r^{(k)}\}$, $\mathcal{S}_k^{\pm} = \{r \in [n_k] \mid \mathbf{l}_r^{(k)} < 0 < \mathbf{u}_r^{(k)}\}$, and $\mathcal{S}_k^- = \{r \in [n_k] \mid \mathbf{l}_r^{(k)} \leq \mathbf{u}_r^{(k)} \leq 0\}$. For neurons in each partitioned set, we define corresponding upper bound $h_{U,r}^{(k)}$ and lower bound $h_{L,r}^{(k)}$ in terms of $\mathbf{l}_r^{(k)}$ and $\mathbf{u}_r^{(k)}$. As we will see shortly, segmenting the activation functions based on $\mathbf{l}_r^{(k)}$ and $\mathbf{u}_r^{(k)}$ is useful to bound a given activation function. We note there are multiple ways of segmenting the activation functions and defining the partitioned sets (e.g. based on the values of $\mathbf{l}_r^{(k)}, \mathbf{u}_r^{(k)}$ rather than their signs), and we can easily incorporate this into our framework to provide the corresponding explicit output bounds for the new partition sets. In the case study, we consider $\mathcal{S}_k^+, \mathcal{S}_k^{\pm}$ and $\mathcal{S}_k^-$ for the four activations, as this partition reflects the curvature of tanh, sigmoid and arctan functions and activation states of ReLU.

**Bounding tanh/sigmoid/arctan.** For tanh activation, $\sigma(y) = \frac{1-e^{-2y}}{1+e^{-2y}}$; for sigmoid activation, $\sigma(y) = \frac{1}{1+e^{-y}}$; for arctan activation, $\sigma(y) = \arctan(y)$. All functions are convex on one side $(y < 0)$ and concave on the other side $(y > 0)$, thus the same rules can be used to find $h_{U,r}^{(k)}$ and $h_{L,r}^{(k)}$. Below we call $(\mathbf{l}_r^{(k)}, \sigma(\mathbf{l}_r^{(k)}))$ as left end-point and $(\mathbf{u}_r^{(k)}, \sigma(\mathbf{u}_r^{(k)}))$ as right end-point. For $r \in \mathcal{S}_k^+$, since $\sigma(y)$ is concave, we can let $h_{U,r}^{(k)}$ be any tangent line of $\sigma(y)$ at point $d \in [\mathbf{l}_r^{(k)}, \mathbf{u}_r^{(k)}]$, and let $h_{L,r}^{(k)}$ pass the two end-points. Similarly, $\sigma(y)$ is concave for $r \in \mathcal{S}_k^+$, thus we can let $h_{L,r}^{(k)}$ be any tangent line of $\sigma(y)$ at point $d \in [\mathbf{l}_r^{(k)}, \mathbf{u}_r^{(k)}]$ and let $h_{U,r}^{(k)}$ pass the two end-points. Lastly, for $r \in \mathcal{S}_k^{\pm}$, we can let $h_{U,r}^{(k)}$ be the tangent line that passes the left end-point and $(d, \sigma(d))$ where $d \geq 0$ and $h_{U,r}^{(k)}$ be the tangent line that passes the right end-point and $(d, \sigma(d))$ where $d \leq 0$. The value of $d$ for transcendental functions can be found using a binary search. The plots of upper and lower bounds for tanh and sigmoid are in Figure 1 and 3 (in Appendix). Plots for arctan are similar and so omitted.

**Bounding ReLU.** For ReLU activation, $\sigma(y) = \max(0, y)$. If $r \in \mathcal{S}_k^+$, we have $\sigma(y) = y$ and so we can set $h_{U,r}^{(k)} = h_{L,r}^{(k)} = y$; if $r \in \mathcal{S}_k^-$, we have $\sigma(y) = 0$, and thus we can set $h_{U,r}^{(k)} = h_{L,r}^{(k)} = 0$; if $r \in \mathcal{S}_k^{\pm}$, we can set $h_{U,r}^{(k)} = \frac{\mathbf{u}_r^{(k)}}{\mathbf{u}_r^{(k)} - \mathbf{l}_r^{(k)}}(y - \mathbf{l}_r^{(k)})$ and $h_{L,r}^{(k)} = ay$, $0 \leq a \leq 1$. Setting $a = \frac{\mathbf{u}_r^{(k)}}{\mathbf{u}_r^{(k)} - \mathbf{l}_r^{(k)}}$ leads to the linear lower bound used in Fast-Lin [20]. Thus, Fast-Lin is a special case under our framework. We propose to *adaptively* choose $a$, where we set $a = 1$ when $\mathbf{u}_r^{(k)} \geq |\mathbf{l}_r^{(k)}|$ and $a = 0$ when $\mathbf{u}_r^{(k)} < |\mathbf{l}_r^{(k)}|$. In this way, the area between the lower bound $h_{L,r}^{(k)} = ay$ and $\sigma(y)$ (which reflects the gap between the lower bound and the ReLU function) is always minimized. As shown in our experiments, the adaptive selection of $h_{L,r}^{(k)}$ based on the value of $\mathbf{u}_r^{(k)}$ and $\mathbf{l}_r^{(k)}$ helps to achieve a tighter certified lower bound. Figure 4 (in Appendix) illustrates the idea discussed here.

**Summary.** We summarized the above analysis on choosing valid linear functions $h_{U,r}^{(k)}$ and $h_{L,r}^{(k)}$ in Table 2 and 3. In general, as long as $h_{U,r}^{(k)}$ and $h_{L,r}^{(k)}$ are identified for the activation functions, we can use Corollary 3.3 to compute certified lower bounds for general activation functions. Note that there remain many other choices of $h_{U,r}^{(k)}$ and $h_{L,r}^{(k)}$ as valid upper/lower bounds of $\sigma(y)$, but ideally, we would like them to be close to $\sigma(y)$ in order to achieve a tighter lower bound of minimum distortion.

### 3.3 Extension to quadratic bounds

In addition to the linear bounds on activation functions, the proposed CROWN framework can also incorporate quadratic bounds by adding a quadratic term to $h_{U,r}^{(k)}$ and $h_{L,r}^{(k)}$: $h_{U,r}^{(k)}(y) = \eta_{U,r}^{(k)} y^2 + \alpha_{U,r}^{(k)}(y + \beta_{U,r}^{(k)})$, $h_{L,r}^{(k)}(y) = \eta_{L,r}^{(k)} y^2 + \alpha_{L,r}^{(k)}(y + \beta_{L,r}^{(k)})$, where $\eta_{U,r}^{(k)}, \eta_{L,r}^{(k)} \in \mathbb{R}$. Following the procedure of unwrapping the activation functions at the layer $m - 1$, we show in Appendix D that the output upper bound and lower bound with quadratic approximations are:

$$f_j^U(\mathbf{x}) = \Phi_{m-2}(\mathbf{x})^\top \mathbf{Q}_U^{(m-1)} \Phi_{m-2}(\mathbf{x}) + 2\mathbf{p}_U^{(m-1)} \Phi_{m-2}(\mathbf{x}) + s_U^{(m-1)}, \qquad (3)$$

$$f_j^L(\mathbf{x}) = \Phi_{m-2}(\mathbf{x})^\top \mathbf{Q}_L^{(m-1)} \Phi_{m-2}(\mathbf{x}) + 2\mathbf{p}_L^{(m-1)} \Phi_{m-2}(\mathbf{x}) + s_L^{(m-1)}, \qquad (4)$$

where $\mathbf{Q}_U^{(m-1)} = \mathbf{W}^{(m-1)\top} \mathbf{D}_U^{(m-1)} \mathbf{W}^{(m-1)}$, $\mathbf{Q}_L^{(m-1)} = \mathbf{W}^{(m-1)\top} \mathbf{D}_L^{(m-1)} \mathbf{W}^{(m-1)}$, $\mathbf{p}_U^{(m-1)}$, $\mathbf{p}_L^{(m-1)}$, $s_U^{(m-1)}$, and $s_L^{(m-1)}$ are defined in Appendix D due to page limit. When $m = 2$, $\Phi_{m-2}(\mathbf{x}) = \mathbf{x}$ and we can directly optimize over $\mathbf{x} \in \mathbb{B}_p(\mathbf{x_0}, \epsilon)$; otherwise, we can use the post activation bounds of layer $m - 2$ as the constraints. $\mathbf{D}_U^{(m-1)}$ in (3) is a diagonal matrix with $i$-th entry being $\mathbf{W}_{j,i}^{(m)} \eta_{U,i}^{(m-1)}$, if $\mathbf{W}_{j,i}^{(m)} \geq 0$ or $\mathbf{W}_{j,i}^{(m)} \eta_{L,i}^{(m-1)}$, if $\mathbf{W}_{j,i}^{(m)} < 0$. Thus, in general $\mathbf{Q}_U^{(m-1)}$ is indefinite, resulting in a non-convex optimization when finding the global bounds as in Corollary 3.3. Fortunately, by properly choosing the quadratic bounds, we can make the problem $\max_{\mathbf{x} \in \mathbb{B}_p(\mathbf{x_0}, \epsilon)} f_j^U(\mathbf{x})$ into a convex Quadratic Programming problem; for example, we can let $\eta_{U,i}^{(m-1)} = 0$ for all $\mathbf{W}_{j,i}^{(m)} > 0$ and let $\eta_{L,i}^{(m-1)} > 0$ to make $\mathbf{D}_U^{(m-1)}$ have only negative and zero diagonals for the maximization problem – this is equivalent to applying a linear upper bound and a quadratic lower bound to bound the activation function. Similarly, for $\mathbf{D}_L^{(m-1)}$, we let $\eta_{U,i}^{(m-1)} = 0$ for all $\mathbf{W}_{j,i}^{(m)} < 0$ and let $\eta_{L,i}^{(m-1)} > 0$ to make $\mathbf{D}_L^{(m-1)}$ have non-negative diagonals and hence the problem $\min_{\mathbf{x} \in \mathbb{B}_p(\mathbf{x_0}, \epsilon)} f_j^L(\mathbf{x})$ is convex. We can solve this convex program with projected gradient descent (PGD) for $\mathbf{x} \in \mathbb{B}_p(\mathbf{x_0}, \epsilon)$ and Armijo line search. Empirically, we find that PGD usually converges within a few iterations.

## 4 Experiments

**Methods.** For ReLU networks, CROWN-Ada is CROWN with adaptive linear bounds (Sec. 3.2), CROWN-Quad is CROWN with quadratic bounds (Sec. 3.3). Fast-Lin and Fast-Lip are state-of-the-art fast certified lower bound proposed in [20]. Reluplex can solve the exact minimum adversarial distortion but is only computationally feasible for very small networks. LP-Full is based on the LP formulation in [18] and we solve LPs for each neuron exactly to achieve the best possible bound. For networks with other activation functions, CROWN-general is our proposed method.

**Model and Dataset.** We evaluate CROWN and other baselines on multi-layer perceptron (MLP) models trained on MNIST and CIFAR-10 datasets. We denote a feed-forward network with $m$ layers and $n$ neurons per layer as $m \times [n]$. For models with ReLU activation, we use pretrained models provided by [20] and also evaluate the same set of 100 random test images and random attack targets as in [20] (according to their released code) to make our results comparable. For training NN models with other activation functions, we search for best learning rate and weight decay parameters to achieve a similar level of accuracy as ReLU models.

**Implementation and Setup.** We implement our algorithm using Python (numpy with numba). Most computations in our method are matrix operations that can be automatically parallelized by the BLAS library; however, we set the number of BLAS threads to 1 for a fair comparison to other methods. Experiments were conducted on an Intel Skylake server CPU running at 2.0 GHz on Google Cloud. Our code is available at `https://github.com/CROWN-Robustness/Crown`

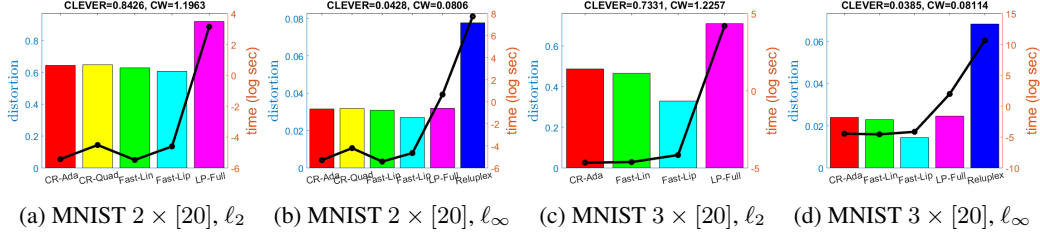

(a) MNIST $2 \times [20]$, $\ell_2$  (b) MNIST $2 \times [20]$, $\ell_\infty$  (c) MNIST $3 \times [20]$, $\ell_2$  (d) MNIST $3 \times [20]$, $\ell_\infty$

Figure 2: Certified lower bounds and minimum distortion comparisons for $\ell_2$ and $\ell_\infty$ distortions. Left y-axis is distortion and right y-axis (black line) is computation time (seconds, logarithmic scale). On the top of figures are the avg. CLEVER score and the upper bound found by C&W attack [6]. From left to right in (a)-(d): CROWN-Ada, (CROWN-Quad), Fast-Lin, Fast-Lip, LP-Full and (Reluplex).

Table 4: Comparison of certified lower bounds on large ReLU networks. Bounds are the average over 100 images (skipped misclassified images) with random attack targets. Percentage improvements are calculated against Fast-Lin as Fast-Lip is worse than Fast-Lin.

| Network | | Certified Bounds | | | Improvement (%) | Average Computation Time (sec) | | |
|---|---|---|---|---|---|---|---|---|
| | $\ell_p$ norm | Fast-Lin | Fast-Lip | CROWN-Ada | CROWN-Ada vs Fast-Lin | Fast-Lin | Fast-Lip | CROWN-Ada |
| MNIST $4 \times [1024]$ | $\ell_1$ | 1.57649 | 0.72800 | **1.88217** | +19% | 1.80 | 2.04 | 3.54 |
| | $\ell_2$ | 0.18891 | 0.06487 | **0.22811** | +21% | 1.78 | 1.96 | 3.79 |
| | $\ell_\infty$ | 0.00823 | 0.00264 | **0.00997** | +21% | 1.53 | 2.17 | 3.57 |
| CIFAR-10 $7 \times [1024]$ | $\ell_1$ | 0.86468 | 0.09239 | **1.09067** | +26% | 13.21 | 19.76 | 22.43 |
| | $\ell_2$ | 0.05937 | 0.00407 | **0.07496** | +26% | 12.57 | 18.71 | 21.82 |
| | $\ell_\infty$ | 0.00134 | 0.00008 | **0.00169** | +26% | 8.98 | 20.34 | 16.66 |

Table 5: Comparison of certified lower bounds by CROWN-Ada on ReLU networks and CROWN-general on networks with tanh, sigmoid and arctan activations. CIFAR models with sigmoid activations achieve much worse accuracy than other networks and are thus excluded.

| Network | | Certified Bounds by CROWN-Ada and CROWN-general | | | | Average Computation Time (sec) | | | |
|---|---|---|---|---|---|---|---|---|---|
| | $\ell_p$ norm | ReLU | tanh | sigmoid | arctan | ReLU | tanh | sigmoid | arctan |
| MNIST $3 \times [1024]$ | $\ell_1$ | 3.00231 | 2.48407 | 2.94239 | 2.33246 | 1.25 | 1.61 | 1.68 | 1.70 |
| | $\ell_2$ | 0.50841 | 0.27287 | 0.44471 | 0.30345 | 1.26 | 1.76 | 1.61 | 1.75 |
| | $\ell_\infty$ | 0.02576 | 0.01182 | 0.02122 | 0.01363 | 1.37 | 1.78 | 1.76 | 1.77 |
| CIFAR-10 $6 \times [2048]$ | $\ell_1$ | 0.91201 | 0.44059 | - | 0.46198 | 71.62 | 89.77 | - | 83.80 |
| | $\ell_2$ | 0.05245 | 0.02538 | - | 0.02515 | 71.51 | 84.22 | - | 83.12 |
| | $\ell_\infty$ | 0.00114 | 0.00055 | - | 0.00055 | 49.28 | 59.72 | - | 58.04 |

**Results on Small Networks.** Figure 2 shows the certified lower bound for $\ell_2$ and $\ell_\infty$ distortions found by different algorithms on small networks, where Reluplex is feasible and we can observe the gap between different certified lower bounds and the true minimum adversarial distortion. Reluplex and LP-Full are orders of magnitudes slower than other methods (note the logarithmic scale on right y-axis), and CROWN-Quad (for 2-layer) and CROWN-Ada achieve the largest lower bounds. Improvements of CROWN-Ada over Fast-Lin are more significant in larger NNs, as we show below.

**Results on Large ReLU Networks.** Table 4 demonstrates the lower bounds found by different algorithms for all common $\ell_p$ norms. CROWN-Ada significantly outperforms Fast-Lin and Fast-Lip, while the computation time increased by less than 2X over Fast-Lin, and is comparable with Fast-Lip. See Appendix for results on more networks.

**Results on Different Activations.** Table 7 compares the certified lower bound computed by CROWN-general for four activation functions and different $\ell_p$ norm on large networks. CROWN-general is able to certify non-trivial lower bounds for all four activation functions efficiently. Comparing to CROWN-Ada on ReLU networks, certifying general activations that are not piece-wise linear only incurs about 20% additional computational overhead.

## 5 Conclusion

We have presented a general framework CROWN to efficiently compute a certified lower bound of minimum distortion in neural networks for any given data point $\mathbf{x_0}$. CROWN features adaptive bounds for improved robustness certification and applies to general activation functions. Moreover, experiments show that (1) CROWN outperforms state-of-the-art baselines on ReLU networks and (2) CROWN can efficiently certify non-trivial lower bounds for large networks with over 10K neurons and with different activation functions.

## Acknowledgement

This work was supported in part by NSF IIS-1719097, Intel faculty award, Google Cloud Credits for Research Program and GPUs donated by NVIDIA. Tsui-Wei Weng and Luca Daniel are partially supported by MIT-IBM Watson AI Lab and MIT-Skoltech program.

## Footnotes

*Work done during internship at IBM Research.   †Equal contribution.

[2]The bound can be further improved by considering $g(\mathbf{x}) := f_c(\mathbf{x}) - f_t(\mathbf{x})$ and replacing the last layer's weights by $\mathbf{W}_{c,:}^{(m)} - \mathbf{W}_{t,:}^{(m)}$. This is also used by [20].

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
