[Supplementary Material]

# A Proof of Theorem 3.2

Given an $m$-layer neural network function $f : \mathbb{R}^{n_0} \to \mathbb{R}^{n_m}$ with pre-activation bounds $\mathbf{l}^{(k)}$ and $\mathbf{u}^{(k)}$ for $\mathbf{x} \in \mathbb{B}_p(\mathbf{x_0}, \epsilon)$ and $\forall k \in [m-1]$, let the pre-activation inputs for the $i$-th neuron at layer $m-1$ be $\mathbf{y}_i^{(m-1)} := \mathbf{W}_{i,:}^{(m-1)} \Phi_{m-2}(\mathbf{x}) + \mathbf{b}_i^{(m-1)}$. The $j$-th output of the neural network is the following:

$$f_j(\mathbf{x}) = \sum_{i=1}^{n_{m-1}} \mathbf{W}_{j,i}^{(m)} [\Phi_{m-1}(\mathbf{x})]_i + \mathbf{b}_j^{(m)}, \tag{5}$$

$$= \sum_{i=1}^{n_{m-1}} \mathbf{W}_{j,i}^{(m)} \sigma(\mathbf{y}_i^{(m-1)}) + \mathbf{b}_j^{(m)},$$

$$= \underbrace{\sum_{\mathbf{W}_{j,i}^{(m)} \geq 0} \mathbf{W}_{j,i}^{(m)} \sigma(\mathbf{y}_i^{(m-1)})}_{F_1} + \underbrace{\sum_{\mathbf{W}_{j,i}^{(m)} < 0} \mathbf{W}_{j,i}^{(m)} \sigma(\mathbf{y}_i^{(m-1)})}_{F_2} + \mathbf{b}_j^{(m)}. \tag{6}$$

Assume the activation function $\sigma(y)$ is bounded by two linear functions $h_{U,i}^{(m-1)}, h_{L,i}^{(m-1)}$ in Definition 3.1, we have

$$h_{L,i}^{(m-1)}(\mathbf{y}_i^{(m-1)}) \leq \sigma(\mathbf{y}_i^{(m-1)}) \leq h_{U,i}^{(m-1)}(\mathbf{y}_i^{(m-1)}).$$

Thus, if the associated weight $\mathbf{W}_{j,i}^{(m)}$ to the $i$-th neuron is non-negative (the terms in $F_1$ bracket), we have

$$\mathbf{W}_{j,i}^{(m)} \cdot h_{L,i}^{(m-1)}(\mathbf{y}_i^{(m-1)}) \leq \mathbf{W}_{j,i}^{(m)} \sigma(\mathbf{y}_i^{(m-1)}) \leq \mathbf{W}_{j,i}^{(m)} \cdot h_{U,i}^{(m-1)}(\mathbf{y}_i^{(m-1)}); \tag{7}$$

otherwise (the terms in $F_2$ bracket), we have

$$\mathbf{W}_{j,i}^{(m)} \cdot h_{U,i}^{(m-1)}(\mathbf{y}_i^{(m-1)}) \leq \mathbf{W}_{j,i}^{(m)} \sigma(\mathbf{y}_i^{(m-1)}) \leq \mathbf{W}_{j,i}^{(m)} \cdot h_{L,i}^{(m-1)}(\mathbf{y}_i^{(m-1)}). \tag{8}$$

**Upper bound.** Let $f_j^{U,m-1}(\mathbf{x})$ be an upper bound of $f_j(\mathbf{x})$. To compute $f_j^{U,m-1}(\mathbf{x})$, (6), (7) and (8) are the key equations. Precisely, for the $\mathbf{W}_{j,i}^{(m)} \geq 0$ terms in (6), the upper bound is the right-hand-side (RHS) in (7); and for the $\mathbf{W}_{j,i}^{(m)} < 0$ terms in (6), the upper bound is the RHS in (8). Thus, we obtain:

$$f_j^{U,m-1}(\mathbf{x})$$

$$= \sum_{\mathbf{W}_{j,i}^{(m)} \geq 0} \mathbf{W}_{j,i}^{(m)} \cdot h_{U,i}^{(m-1)}(\mathbf{y}_i^{(m-1)}) + \sum_{\mathbf{W}_{j,i}^{(m)} < 0} \mathbf{W}_{j,i}^{(m)} \cdot h_{L,i}^{(m-1)}(\mathbf{y}_i^{(m-1)}) + \mathbf{b}_j^{(m)}, \tag{9}$$

$$= \sum_{\mathbf{W}_{j,i}^{(m)} \geq 0} \mathbf{W}_{j,i}^{(m)} \alpha_{U,i}^{(m-1)}(\mathbf{y}_i^{(m-1)} + \beta_{U,i}^{(m-1)}) + \sum_{\mathbf{W}_{j,i}^{(m)} < 0} \mathbf{W}_{j,i}^{(m)} \alpha_{L,i}^{(m-1)}(\mathbf{y}_i^{(m-1)} + \beta_{L,i}^{(m-1)}) + \mathbf{b}_j^{(m)}, \tag{10}$$

$$= \sum_{i=1}^{n_{m-1}} \mathbf{W}_{j,i}^{(m)} \lambda_{j,i}^{(m-1)}(\mathbf{y}_i^{(m-1)} + \mathbf{\Delta}_{i,j}^{(m-1)}) + \mathbf{b}_j^{(m)}, \tag{11}$$

$$= \sum_{i=1}^{n_{m-1}} \mathbf{\Lambda}_{j,i}^{(m-1)} \left( \sum_{r=1}^{n_{m-2}} \mathbf{W}_{i,r}^{(m-1)} [\Phi_{m-2}(\mathbf{x})]_r + \mathbf{b}_i^{(m-1)} + \mathbf{\Delta}_{i,j}^{(m-1)} \right) + \mathbf{b}_j^{(m)}, \tag{12}$$

$$= \sum_{i=1}^{n_{m-1}} \mathbf{\Lambda}_{j,i}^{(m-1)} \left( \sum_{r=1}^{n_{m-2}} \mathbf{W}_{i,r}^{(m-1)} [\Phi_{m-2}(\mathbf{x})]_r \right) + \sum_{i=1}^{n_{m-1}} \mathbf{\Lambda}_{j,i}^{(m-1)} (\mathbf{b}_i^{(m-1)} + \mathbf{\Delta}_{i,j}^{(m-1)}) + \mathbf{b}_j^{(m)}, \tag{13}$$

$$= \sum_{r=1}^{n_{m-2}} \left( \sum_{i=1}^{n_{m-1}} \mathbf{\Lambda}_{j,i}^{(m-1)} \mathbf{W}_{i,r}^{(m-1)} \right) [\Phi_{m-2}(\mathbf{x})]_r + \left( \sum_{i=1}^{n_{m-1}} \mathbf{\Lambda}_{j,i}^{(m-1)} (\mathbf{b}_i^{(m-1)} + \mathbf{\Delta}_{i,j}^{(m-1)}) + \mathbf{b}_j^{(m)} \right), \tag{14}$$

$$= \sum_{r=1}^{n_{m-2}} \tilde{\mathbf{W}}_{j,r}^{(m-1)} [\Phi_{m-2}(\mathbf{x})]_r + \tilde{\mathbf{b}}_j^{(m-1)}. \tag{15}$$

From (9) to (10), we replace $h_{U,i}^{(m-1)}(\mathbf{y}_i^{(m-1)})$ and $h_{L,i}^{(m-1)}(\mathbf{y}_i^{(m-1)})$ by their definitions; from (10) to (11), we use variables $\lambda_{j,i}^{(m-1)}$ and $\boldsymbol{\Delta}_{j,i}^{(m-1)}$ to denote the slopes in front of $\mathbf{y}_i^{(m-1)}$ and the intercepts in the parentheses:

$$\lambda_{j,i}^{(m-1)} = \begin{cases} \alpha_{U,i}^{(m-1)} & \text{if } \mathbf{W}_{j,i}^{(m)} \geq 0 \quad ( \iff \boldsymbol{\Lambda}_{j,:}^{(m)}\mathbf{W}_{:,i}^{(m)} \geq 0); \\ \alpha_{L,i}^{(m-1)} & \text{if } \mathbf{W}_{j,i}^{(m)} < 0 \quad ( \iff \boldsymbol{\Lambda}_{j,:}^{(m)}\mathbf{W}_{:,i}^{(m)} < 0); \end{cases} \tag{16}$$

$$\boldsymbol{\Delta}_{i,j}^{(m-1)} = \begin{cases} \beta_{U,i}^{(m-1)} & \text{if } \mathbf{W}_{j,i}^{(m)} \geq 0 \quad ( \iff \boldsymbol{\Lambda}_{j,:}^{(m)}\mathbf{W}_{:,i}^{(m)} \geq 0); \\ \beta_{L,i}^{(m-1)} & \text{if } \mathbf{W}_{j,i}^{(m)} < 0 \quad ( \iff \boldsymbol{\Lambda}_{j,:}^{(m)}\mathbf{W}_{:,i}^{(m)} < 0). \end{cases} \tag{17}$$

From (11) to (12), we replace $\mathbf{y}_i^{(m-1)}$ with its definition and let $\boldsymbol{\Lambda}_{j,i}^{(m-1)} := \mathbf{W}_{j,i}^{(m)}\lambda_{j,i}^{(m-1)}$. We further let $\boldsymbol{\Lambda}_{j,:}^{(m)} = \mathbf{e}_j^\top$ (the standard unit vector with the only non-zero $j$th element equal to 1), and thus we can rewrite the conditions of $\mathbf{W}_{j,i}^{(m)}$ in (16) and (17) as $\boldsymbol{\Lambda}_{j,:}^{(m)}\mathbf{W}_{:,i}^{(m)}$. From (12) to (13), we collect the constant terms that are not related to $\mathbf{x}$. From (13) to (14), we swap the summation order of $i$ and $r$, and the coefficients in front of $[\Phi_{m-2}(x)]_r$ can be combined into a new equivalent weight $\tilde{\mathbf{W}}_{j,r}^{(m-1)}$ and the constant term can combined into a new equivalent bias $\tilde{\mathbf{b}}_j^{(m-1)}$ in (15):

$$\tilde{\mathbf{W}}_{j,r}^{(m-1)} = \sum_{i=1}^{n_{m-1}} \boldsymbol{\Lambda}_{j,i}^{(m-1)}\mathbf{W}_{i,r}^{(m-1)} = \boldsymbol{\Lambda}_{j,:}^{(m-1)}\mathbf{W}_{:,r}^{(m-1)},$$

$$\tilde{\mathbf{b}}_j^{(m-1)} = \sum_{i=1}^{n_{m-1}} \boldsymbol{\Lambda}_{j,i}^{(m-1)}(\mathbf{b}_i^{(m-1)} + \boldsymbol{\Delta}_{i,j}^{(m-1)}) + \mathbf{b}_j^{(m)} = \boldsymbol{\Lambda}_{j,:}^{(m-1)}(\mathbf{b}^{(m-1)} + \boldsymbol{\Delta}_{:,j}^{(m-1)}) + \mathbf{b}_j^{(m)}.$$

Notice that after defining the new equivalent weight $\tilde{\mathbf{W}}_{j,r}^{(m-1)}$ and equivalent bias $\tilde{\mathbf{b}}_j^{(m-1)}$, $f_j^{U,m-1}(\mathbf{x})$ in (15) and $f_j(\mathbf{x})$ in (5) are in the same form. Thus, we can repeat the above procedure again to obtain an upper bound of $f_j^{U,m-1}(\mathbf{x})$, i.e. $f_j^{U,m-2}(\mathbf{x})$:

$$\boldsymbol{\Lambda}_{j,i}^{(m-2)} = \tilde{\mathbf{W}}_{j,i}^{(m-1)}\lambda_{j,i}^{(m-2)}$$
$$= \boldsymbol{\Lambda}_{j,:}^{(m-1)}\mathbf{W}_{:,i}^{(m-1)}\lambda_{j,i}^{(m-2)}$$
$$\tilde{\mathbf{W}}_{j,r}^{(m-2)} = \boldsymbol{\Lambda}_{j,:}^{(m-2)}\mathbf{W}_{:,r}^{(m-2)}$$
$$\tilde{\mathbf{b}}_j^{(m-2)} = \boldsymbol{\Lambda}_{j,:}^{(m-2)}(\mathbf{b}^{(m-2)} + \boldsymbol{\Delta}_{:,j}^{(m-2)}) + \tilde{\mathbf{b}}_j^{(m-1)}$$

$$\lambda_{j,i}^{(m-2)} = \begin{cases} \alpha_{U,i}^{(m-2)} & \text{if } \tilde{\mathbf{W}}_{j,i}^{(m-1)} \geq 0 \quad ( \iff \boldsymbol{\Lambda}_{j,:}^{(m-1)}\mathbf{W}_{:,i}^{(m-1)} \geq 0); \\ \alpha_{L,i}^{(m-2)} & \text{if } \tilde{\mathbf{W}}_{j,i}^{(m-1)} < 0 \quad ( \iff \boldsymbol{\Lambda}_{j,:}^{(m-1)}\mathbf{W}_{:,i}^{(m-1)} < 0); \end{cases}$$

$$\boldsymbol{\Delta}_{i,j}^{(m-2)} = \begin{cases} \beta_{U,i}^{(m-2)} & \text{if } \tilde{\mathbf{W}}_{j,i}^{(m-1)} \geq 0 \quad ( \iff \boldsymbol{\Lambda}_{j,:}^{(m-1)}\mathbf{W}_{:,i}^{(m-1)} \geq 0); \\ \beta_{L,i}^{(m-2)} & \text{if } \tilde{\mathbf{W}}_{j,i}^{(m-1)} < 0 \quad ( \iff \boldsymbol{\Lambda}_{j,:}^{(m-1)}\mathbf{W}_{:,i}^{(m-1)} < 0). \end{cases}$$

and repeat again iteratively until obtain the final upper bound $f_j^{U,1}(\mathbf{x})$, where $f_j(\mathbf{x}) \leq f_j^{U,m-1}(\mathbf{x}) \leq f_j^{U,m-2}(\mathbf{x}) \leq \ldots \leq f_j^{U,1}(\mathbf{x})$. We let $f_j(\mathbf{x})$ denote the final upper bound $f_j^{U,1}(\mathbf{x})$, and we have

$$f_j^U(\mathbf{x}) = \boldsymbol{\Lambda}_{j,:}^{(0)}\mathbf{x} + \sum_{k=1}^m \boldsymbol{\Lambda}_{j,:}^{(k)}(\mathbf{b}^{(k)} + \boldsymbol{\Delta}_{:,j}^{(k)})$$

and ($\odot$ is the Hadamard product)

$$\boldsymbol{\Lambda}_{j,:}^{(k-1)} = \begin{cases} \mathbf{e}_j^\top & \text{if } k = m+1; \\ (\boldsymbol{\Lambda}_{j,:}^{(k)}\mathbf{W}^{(k)}) \odot \lambda_{j,:}^{(k-1)} & \text{if } k \in [m]. \end{cases}$$

and $\forall i \in [n_k]$,

$$\lambda_{j,i}^{(k)} = \begin{cases} \alpha_{U,i}^{(k)} & \text{if } k \in [m-1], \, \boldsymbol{\Lambda}_{j,:}^{(k+1)}\mathbf{W}_{:,i}^{(k+1)} \geq 0; \\ \alpha_{L,i}^{(k)} & \text{if } k \in [m-1], \, \boldsymbol{\Lambda}_{j,:}^{(k+1)}\mathbf{W}_{:,i}^{(k+1)} < 0; \\ 1 & \text{if } k = 0. \end{cases}$$

$$\mathbf{\Delta}_{i,j}^{(k)} = \begin{cases} \beta_{U,i}^{(k)} & \text{if } k \in [m-1],\ \mathbf{\Lambda}_{j,:}^{(k+1)}\mathbf{W}_{:,i}^{(k+1)} \geq 0; \\ \beta_{L,i}^{(k)} & \text{if } k \in [m-1],\ \mathbf{\Lambda}_{j,:}^{(k+1)}\mathbf{W}_{:,i}^{(k+1)} < 0; \\ 0 & \text{if } k = m. \end{cases}$$

**Lower bound.** The above derivations of upper bound can be applied similarly to deriving lower bounds of $f_j(\mathbf{x})$, and the only difference is now we need to use the LHS of (7) and (8) (rather than RHS when deriving upper bound) to bound the two terms in (6). Thus, following the same procedure in deriving the upper bounds, we can iteratively unwrap the activation functions and obtain a final lower bound $f_j^{L,1}(\mathbf{x})$, where $f_j(\mathbf{x}) \geq f_j^{L,m-1}(\mathbf{x}) \geq f_j^{L,m-2}(\mathbf{x}) \geq \ldots \geq f_j^{L,1}(\mathbf{x})$. Let $f_j^L(\mathbf{x}) = f_j^{L,1}(\mathbf{x})$, we have:

$$f_j^L(\mathbf{x}) = \mathbf{\Omega}_{j,:}^{(0)}\mathbf{x} + \sum_{k=1}^{m} \mathbf{\Omega}_{j,:}^{(k)}(\mathbf{b}^{(k)} + \mathbf{\Theta}_{:,j}^{(k)})$$

$$\mathbf{\Omega}_{j,:}^{(k-1)} = \begin{cases} \mathbf{e}_j^\top & \text{if } k = m+1; \\ (\mathbf{\Omega}_{j,:}^{(k)}\mathbf{W}^{(k)}) \odot \omega_{j,:}^{(k-1)} & \text{if } k \in [m]. \end{cases}$$

and $\forall i \in [n_k]$,

$$\omega_{j,i}^{(k)} = \begin{cases} \alpha_{L,i}^{(k)} & \text{if } k \in [m-1],\ \mathbf{\Omega}_{j,:}^{(k+1)}\mathbf{W}_{:,i}^{(k+1)} \geq 0; \\ \alpha_{U,i}^{(k)} & \text{if } k \in [m-1],\ \mathbf{\Omega}_{j,:}^{(k+1)}\mathbf{W}_{:,i}^{(k+1)} < 0; \\ 1 & \text{if } k = 0. \end{cases}$$

$$\mathbf{\Theta}_{i,j}^{(k)} = \begin{cases} \beta_{L,i}^{(k)} & \text{if } k \in [m-1],\ \mathbf{\Omega}_{j,:}^{(k+1)}\mathbf{W}_{:,i}^{(k+1)} \geq 0; \\ \beta_{U,i}^{(k)} & \text{if } k \in [m-1],\ \mathbf{\Omega}_{j,:}^{(k+1)}\mathbf{W}_{:,i}^{(k+1)} < 0; \\ 0 & \text{if } k = m. \end{cases}$$

Indeed, $\lambda_{j,i}^{(k)}$ and $\omega_{j,i}^{(k)}$ only differs in the conditions of selecting $\alpha_{U,i}^{(k)}$ or $\alpha_{L,i}^{(k)}$; similarly for $\mathbf{\Delta}_{i,j}^{(k)}$ and $\mathbf{\Theta}_{i,j}^{(k)}$.

## B Proof of Corollary 3.3

**Definition B.1** (Dual norm). *Let $\|\cdot\|$ be a norm on $\mathbb{R}^n$. The associated dual norm, denoted as $\|\cdot\|_*$, is defined as*

$$\|\mathbf{a}\|_* = \{\sup_{\mathbf{y}} \mathbf{a}^\top\mathbf{y} \mid \|\mathbf{y}\| \leq 1\}.$$

**Global upper bound.** Our goal is to find a *global* upper and lower bound for the $m$-th layer network output $f_j(\mathbf{x}), \forall \mathbf{x} \in \mathbb{B}_p(\mathbf{x_0}, \epsilon)$. By Theorem 3.2, for $\mathbf{x} \in \mathbb{B}_p(\mathbf{x_0}, \epsilon)$, we have $f_j^L(\mathbf{x}) \leq f_j(\mathbf{x}) \leq f_j^U(\mathbf{x})$ and $f_j^U(\mathbf{x}) = \mathbf{\Lambda}_{j,:}^{(0)}\mathbf{x} + \sum_{k=1}^{m} \mathbf{\Lambda}_{j,:}^{(k)}(\mathbf{b}^{(k)} + \mathbf{\Delta}_{:,j}^{(k)})$. Thus define $\gamma_j^U := \max_{\mathbf{x} \in \mathbb{B}_p(\mathbf{x_0}, \epsilon)} f_j^U(\mathbf{x})$, and we have

$$f_j(\mathbf{x}) \leq f_j^U(\mathbf{x}) \leq \max_{\mathbf{x} \in \mathbb{B}_p(\mathbf{x_0}, \epsilon)} f_j^U(\mathbf{x}) = \gamma_j^U,$$

since $\forall \mathbf{x} \in \mathbb{B}_p(\mathbf{x_0}, \epsilon)$. In particular,

$$\max_{\mathbf{x} \in \mathbb{B}_p(\mathbf{x_0}, \epsilon)} f_j^U(\mathbf{x}) = \max_{\mathbf{x} \in \mathbb{B}_p(\mathbf{x_0}, \epsilon)} \left[ \mathbf{\Lambda}_{j,:}^{(0)}\mathbf{x} + \sum_{k=1}^{m} \mathbf{\Lambda}_{j,:}^{(k)}(\mathbf{b}^{(k)} + \mathbf{\Delta}_{:,j}^{(k)}) \right]$$

$$= \left[ \max_{\mathbf{x} \in \mathbb{B}_p(\mathbf{x_0}, \epsilon)} \mathbf{\Lambda}_{j,:}^{(0)}\mathbf{x} \right] + \sum_{k=1}^{m} \mathbf{\Lambda}_{j,:}^{(k)}(\mathbf{b}^{(k)} + \mathbf{\Delta}_{:,j}^{(k)}) \tag{18}$$

$$= \epsilon \left[ \max_{\mathbf{y} \in \mathbb{B}_p(\mathbf{0}, 1)} \mathbf{\Lambda}_{j,:}^{(0)}\mathbf{y} \right] + \mathbf{\Lambda}_{j,:}^{(0)}\mathbf{x_0} + \sum_{k=1}^{m} \mathbf{\Lambda}_{j,:}^{(k)}(\mathbf{b}^{(k)} + \mathbf{\Delta}_{:,j}^{(k)}) \tag{19}$$

$$= \epsilon\|\mathbf{\Lambda}_{j,:}^{(0)}\|_q + \mathbf{\Lambda}_{j,:}^{(0)}\mathbf{x_0} + \sum_{k=1}^{m} \mathbf{\Lambda}_{j,:}^{(k)}(\mathbf{b}^{(k)} + \mathbf{\Delta}_{:,j}^{(k)}). \tag{20}$$

From (18) to (19), let $\mathbf{y} := \frac{\mathbf{x}-\mathbf{x_0}}{\epsilon}$, and thus $\mathbf{y} \in \mathbb{B}_p(\mathbf{0}, 1)$. From (19) to (20), apply Definition B.1 and use the fact that $\ell_q$ norm is dual of $\ell_p$ norm for $p, q \in [1, \infty]$.

**Global lower bound.** Similarly, let $\gamma_j^L := \min_{\mathbf{x} \in \mathbb{B}_p(\mathbf{x_0}, \epsilon)} f_j^L(\mathbf{x})$, we have

$$f_j(\mathbf{x}) \geq f_j^L(\mathbf{x}) \geq \min_{\mathbf{x} \in \mathbb{B}_p(\mathbf{x_0}, \epsilon)} f_j^L(\mathbf{x}) = \gamma_j^L.$$

Since $f_j^L(\mathbf{x}) = \mathbf{\Omega}_{j,:}^{(0)} \mathbf{x} + \sum_{k=1}^m \mathbf{\Omega}_{j,:}^{(k)} (\mathbf{b}^{(k)} + \mathbf{\Theta}_{:,j}^{(k)})$, we can derive $\gamma_j^L$ (similar to the derivation of $\gamma_j^U$) below:

$$\min_{\mathbf{x} \in \mathbb{B}_p(\mathbf{x_0}, \epsilon)} f_j^L(\mathbf{x}) = \min_{\mathbf{x} \in \mathbb{B}_p(\mathbf{x_0}, \epsilon)} \left[ \mathbf{\Omega}_{j,:}^{(0)} \mathbf{x} + \sum_{k=1}^m \mathbf{\Omega}_{j,:}^{(k)} (\mathbf{b}^{(k)} + \mathbf{\Theta}_{:,j}^{(k)}) \right]$$

$$= \left[ \min_{\mathbf{x} \in \mathbb{B}_p(\mathbf{x_0}, \epsilon)} \mathbf{\Omega}_{j,:}^{(0)} \mathbf{x} \right] + \sum_{k=1}^m \mathbf{\Omega}_{j,:}^{(k)} (\mathbf{b}^{(k)} + \mathbf{\Theta}_{:,j}^{(k)})$$

$$= -\epsilon \left[ \max_{\mathbf{y} \in \mathbb{B}_p(\mathbf{0}, 1)} -\mathbf{\Omega}_{j,:}^{(0)} \mathbf{y} \right] + \mathbf{\Omega}_{j,:}^{(0)} \mathbf{x_0} + \sum_{k=1}^m \mathbf{\Omega}_{j,:}^{(k)} (\mathbf{b}^{(k)} + \mathbf{\Theta}_{:,j}^{(k)})$$

$$= -\epsilon \|\mathbf{\Omega}_{j,:}^{(0)}\|_q + \mathbf{\Omega}_{j,:}^{(0)} \mathbf{x_0} + \sum_{k=1}^m \mathbf{\Omega}_{j,:}^{(k)} (\mathbf{b}^{(k)} + \mathbf{\Theta}_{:,j}^{(k)}).$$

Thus, we have

$$\text{(global upper bound)} \quad \gamma_j^U = \epsilon \|\mathbf{\Lambda}_{j,:}^{(0)}\|_q + \mathbf{\Lambda}_{j,:}^{(0)} \mathbf{x_0} + \sum_{k=1}^m \mathbf{\Lambda}_{j,:}^{(k)} (\mathbf{b}^{(k)} + \mathbf{\Delta}_{:,j}^{(k)}),$$

$$\text{(global lower bound)} \quad \gamma_j^L = -\epsilon \|\mathbf{\Omega}_{j,:}^{(0)}\|_q + \mathbf{\Omega}_{j,:}^{(0)} \mathbf{x_0} + \sum_{k=1}^m \mathbf{\Omega}_{j,:}^{(k)} (\mathbf{b}^{(k)} + \mathbf{\Theta}_{:,j}^{(k)}),$$

# C  Illustration of linear upper and lower bounds on sigmoid activation function.

(a) $r \in \mathcal{S}_k^+$      (b) $r \in \mathcal{S}_k^-$      (c) $r \in \mathcal{S}_k^\pm$

Figure 3: The linear upper and lower bounds for $\sigma(y) = $ sigmoid

# D  $f_j^U(\mathbf{x})$ and $f_j^L(\mathbf{x})$ by Quadratic approximation

**Upper bound.** Let $f_j^U(\mathbf{x})$ be an upper bound of $f_j(\mathbf{x})$. To compute $f_j^U(\mathbf{x})$ with quadratic approximations, we can still apply (7) and (8) except that $h_{U,r}^{(k)}(y)$ and $h_{L,r}^{(k)}(y)$ are replaced by the following quadratic functions:

$$h_{U,r}^{(k)}(y) = \eta_{U,r}^{(k)} y^2 + \alpha_{U,r}^{(k)} (y + \beta_{U,r}^{(k)}), \ h_{L,r}^{(k)}(y) = \eta_{L,r}^{(k)} y^2 + \alpha_{L,r}^{(k)} (y + \beta_{L,r}^{(k)}).$$

Figure 4: The linear upper and lower bounds for $\sigma(y) = $ ReLU. For the cases (a) and (b), the linear upper bound and lower bound are exactly the function $\sigma(y)$ in the region (grey-shaded). For (c), we plot three out of many choices of lower bound, and they are $h_{L,r}^{(k)}(y) = 0$ (dashed-dotted), $h_{L,r}^{(k)}(y) = y$ (dashed), and $h_{L,r}^{(k)}(y) = \frac{\mathbf{u}_r^{(k)}}{\mathbf{u}_r^{(k)} - \mathbf{l}_r^{(k)}} y$ (dotted).

Therefore,

$$f_j^U(\mathbf{x}) = \sum_{\mathbf{W}_{j,i}^{(m)} \geq 0} \mathbf{W}_{j,i}^{(m)} \cdot h_{U,i}^{(m-1)}(\mathbf{y}_i^{(m-1)}) + \sum_{\mathbf{W}_{j,i}^{(m)} < 0} \mathbf{W}_{j,i}^{(m)} \cdot h_{L,i}^{(m-1)}(\mathbf{y}_i^{(m-1)}) + \mathbf{b}_j^{(m)}, \quad (21)$$

$$= \sum_{i=1}^{n_{m-1}} \mathbf{W}_{j,i}^{(m)} \left( \tau_{j,i}^{(m-1)} \mathbf{y}_i^{(m-1)2} + \lambda_{j,i}^{(m-1)}(\mathbf{y}_i^{(m-1)} + \boldsymbol{\Delta}_{i,j}^{(m-1)}) \right) + \mathbf{b}_j^{(m)}, \quad (22)$$

$$= \mathbf{y}^{(m-1)\top} \text{diag}(\mathbf{q}_{U,j}^{(m-1)}) \mathbf{y}^{(m-1)} + \boldsymbol{\Lambda}_{j,:}^{(m-1)} \mathbf{y}^{(m-1)} + \mathbf{W}_{j,:}^{(m)} \boldsymbol{\Delta}_{:,j}^{(m-1)}, \quad (23)$$

$$= \Phi_{m-2}(\mathbf{x})^\top \mathbf{Q}_U^{(m-1)} \Phi_{m-2}(\mathbf{x}) + 2\mathbf{p}_U^{(m-1)} \Phi_{m-2}(\mathbf{x}) + s_U^{(m-1)}. \quad (24)$$

From (21) to (22), we replace $h_{U,i}^{(m-1)}(\mathbf{y}_i^{(m-1)})$ and $h_{L,i}^{(m-1)}(\mathbf{y}_i^{(m-1)})$ by their definitions and let

$$(\tau_{j,i}^{(m-1)}, \lambda_{j,i}^{(m-1)}, \boldsymbol{\Delta}_{i,j}^{(m-1)}) = \begin{cases} (\eta_{U,i}^{(m-1)}, \alpha_{U,i}^{(m-1)}, \beta_{U,i}^{(m-1)}) & \text{if } \mathbf{W}_{j,i}^{(m)} \geq 0; \\ (\eta_{L,i}^{(m-1)}, \alpha_{L,i}^{(m-1)}, \beta_{L,i}^{(m-1)}) & \text{if } \mathbf{W}_{j,i}^{(m)} < 0. \end{cases}$$

From (22) to (23), we let $\mathbf{q}_{U,j}^{(m-1)} = \mathbf{W}_{j,:}^{(m)} \odot \tau_{j,i}^{(m-1)}$, and write in the matrix form. From (23) to (24), we substitute $\mathbf{y}^{(m-1)}$ by its definition: $\mathbf{y}^{(m-1)} = \mathbf{W}^{(m-1)} \Phi_{(m-2)}(\mathbf{x}) + \mathbf{b}^{(m-1)}$ and then collect the quadratic terms, linear terms and constant terms of $\Phi_{(m-2)}(\mathbf{x})$, where

$$\mathbf{Q}_U^{(m-1)} = \mathbf{W}^{(m-1)\top} \text{diag}(\mathbf{q}_{U,j}^{(m-1)}) \mathbf{W}^{(m-1)},$$

$$\mathbf{p}_U^{(m-1)} = \mathbf{b}^{(m-1)\top} \odot \mathbf{q}_{U,j}^{(m-1)} + \boldsymbol{\Lambda}_{j,:}^{(m-1)},$$

$$s_U^{(m-1)} = \mathbf{p}_U^{(m-1)} \mathbf{b}^{(m-1)} + \mathbf{W}_{j,:}^{(m)} \boldsymbol{\Delta}_{:,j}^{(m-1)}.$$

**Lower bound.** Similar to the above derivation, we can simply swap $h_{U,r}^{(k)}$ and $h_{L,r}^{(k)}$ and obtain lower bound $f_j^L(\mathbf{x})$:

$$f_j^L(\mathbf{x}) = \sum_{\mathbf{W}_{j,i}^{(m)} < 0} \mathbf{W}_{j,i}^{(m)} \cdot h_{U,i}^{(m-1)}(\mathbf{y}_i^{(m-1)}) + \sum_{\mathbf{W}_{j,i}^{(m)} \geq 0} \mathbf{W}_{j,i}^{(m)} \cdot h_{L,i}^{(m-1)}(\mathbf{y}_i^{(m-1)}) + \mathbf{b}_j^{(m)},$$

$$= \Phi_{m-2}(\mathbf{x})^\top \mathbf{Q}_L^{(m-1)} \Phi_{m-2}(\mathbf{x}) + 2\mathbf{p}_L^{(m-1)} \Phi_{m-2}(\mathbf{x}) + s_L^{(m-1)},$$

where

$$\mathbf{Q}_L^{(m-1)} = \mathbf{W}^{(m-1)\top} \text{diag}(\mathbf{q}_{L,j}^{(m-1)}) \mathbf{W}^{(m-1)}, \quad \mathbf{q}_{L,j}^{(m-1)} = \mathbf{W}_{j,:}^{(m)} \odot \nu_{j,i}^{(m-1)}; \quad (25)$$

$$\mathbf{p}_U^{(m-1)} = \mathbf{b}^{(m-1)\top} \odot \mathbf{q}_{U,j}^{(m-1)} + \boldsymbol{\Lambda}_{j,:}^{(m-1)}, \quad \mathbf{p}_L^{(m-1)} = \mathbf{b}^{(m-1)\top} \odot \mathbf{q}_{L,j}^{(m-1)} + \boldsymbol{\Omega}_{j,:}^{(m-1)}; \quad (26)$$

$$s_U^{(m-1)} = \mathbf{p}_U^{(m-1)} \mathbf{b}^{(m-1)} + \mathbf{W}_{j,:}^{(m)} \boldsymbol{\Delta}_{:,j}^{(m-1)}, \quad s_L^{(m-1)} = \mathbf{p}_L^{(m-1)} \mathbf{b}^{(m-1)} + \mathbf{W}_{j,:}^{(m)} \boldsymbol{\Theta}_{:,j}^{(m-1)}, \quad (27)$$

and

$$(\nu_{j,i}^{(m-1)}, \omega_{j,i}^{(m-1)}, \boldsymbol{\Theta}_{i,j}^{(m-1)}) = \begin{cases} (\eta_{L,i}^{(m-1)}, \alpha_{L,i}^{(m-1)}, \beta_{L,i}^{(m-1)}) & \text{if } \mathbf{W}_{j,i}^{(m)} \geq 0; \\ (\eta_{U,i}^{(m-1)}, \alpha_{U,i}^{(m-1)}, \beta_{U,i}^{(m-1)}) & \text{if } \mathbf{W}_{j,i}^{(m)} < 0. \end{cases} \quad (28)$$

# E  Additional Experimental Results

## E.1  Results on CROWN-Ada

Table 6: Comparison of our proposed certified lower bounds for ReLU with adaptive lower bounds (CROWN-Ada), Fast-Lin and Fast-Lip and Op-nrom. LP-full and Reluplex cannot finish within a reasonable amount of time for all the networks reported here. We also include Op-norm, where we directly compute the operator norm (for example, for $p = 2$ it is the spectral norm) for each layer and use their products as a global Lipschitz constant and then compute the robustness lower bound. CLEVER is an estimated robustness lower bound, and attacking algorithms (including CW [6] and EAD [32]) provide upper bounds of the minimum adversarial distortion. For each norm, we consider the robustness against three targeted attack classes: the runner-up class (with the second largest probability), a random class and the least likely class. It is clear that CROWN-Ada notably improves the lower bound comparing to Fast-Lin, especially for larger and deeper networks, the improvements can be up to 28%.

| Networks | | | Lower bounds and upper bounds (Avg.) | | | | | | | Time per Image (Avg.) | | |
|---|---|---|---|---|---|---|---|---|---|---|---|---|
| | | | Lower Bounds (certified) | | | | improvements over | Uncertified | | Lower Bounds | | |
| Config | $p$ | Target | [20] | | [3] | Our algorithm | | [27] | Attacks | [20] | | Our bound |
| | | | Fast-Lin | Fast-Lip | Op norm | CROWN-Ada | Fast-Lin | CLEVER | CW/EAD | Fast-Lin | Fast-Lip | CROWN-Ada |
| MNIST $2 \times [1024]$ | $\infty$ | runner-up | 0.02256 | 0.01802 | 0.00159 | 0.02467 | +9.4% | 0.0447 | 0.0856 | 163 ms | 179 ms | 128 ms |
| | | rand | 0.03083 | 0.02512 | 0.00263 | 0.03353 | +8.8% | 0.0708 | 0.1291 | 176 ms | 213 ms | 166 ms |
| | | least | 0.03854 | 0.03128 | 0.00369 | 0.04221 | +9.5% | 0.0925 | 0.1731 | 176 ms | 251 ms | 143 ms |
| | 2 | runner-up | 0.46034 | 0.42027 | 0.24327 | 0.50110 | +8.9% | 0.8104 | 1.1874 | 154 ms | 184 ms | 110 ms |
| | | rand | 0.63299 | 0.59033 | 0.40201 | 0.68506 | +8.2% | 1.2841 | 1.8779 | 141 ms | 212 ms | 133 ms |
| | | least | 0.79263 | 0.73133 | 0.56509 | 0.86377 | +9.0% | 1.6716 | 2.4556 | 152 ms | 291 ms | 116 ms |
| | 1 | runner-up | 2.78786 | 3.46500 | 0.20601 | 3.01633 | +8.2% | 4.5970 | 9.5295 | 159 ms | 989 ms | 136 ms |
| | | rand | 3.88241 | 5.10000 | 0.35957 | 4.17760 | +7.6% | 7.4186 | 17.259 | 168 ms | 1.15 s | 157 ms |
| | | least | 4.90809 | 6.36600 | 0.48774 | 5.33261 | +8.6% | 9.9847 | 23.933 | 179 ms | 1.37 s | 144 ms |
| MNIST $3 \times [1024]$ | $\infty$ | runner-up | 0.01830 | 0.01021 | 0.00004 | 0.02114 | +15.5% | 0.0509 | 0.1037 | 805 ms | 1.28 s | 1.33 s |
| | | rand | 0.02216 | 0.01236 | 0.00007 | 0.02576 | +16.2% | 0.0717 | 0.1484 | 782 ms | 859 ms | 1.37 s |
| | | least | 0.02432 | 0.01384 | 0.00009 | 0.02835 | +16.6% | 0.0825 | 0.1777 | 792 ms | 684 ms | 1.37 s |
| | 2 | runner-up | 0.35867 | 0.22120 | 0.06626 | 0.41295 | +15.1% | 0.8402 | 1.3513 | 732 ms | 1.06 s | 1.26 s |
| | | rand | 0.43892 | 0.26980 | 0.10233 | 0.50841 | +15.8% | 1.2441 | 2.0387 | 711 ms | 696 ms | 1.26 s |
| | | least | 0.48361 | 0.30147 | 0.13256 | 0.56167 | +16.1% | 1.4401 | 2.4916 | 723 ms | 655 ms | 1.25 s |
| | 1 | runner-up | 2.08887 | 1.80150 | 0.00734 | 2.39443 | +14.6% | 4.8370 | 10.159 | 685 ms | 2.36 s | 1.15 s |
| | | rand | 2.59898 | 2.25950 | 0.01133 | 3.00231 | +15.5% | 7.2177 | 17.796 | 743 ms | 2.69 s | 1.25 s |
| | | least | 2.87560 | 2.50000 | 0.01499 | 3.33231 | +15.9% | 8.3523 | 22.395 | 729 ms | 3.08 s | 1.31 s |
| MNIST $4 \times [1024]$ | $\infty$ | runner-up | 0.00715 | 0.00219 | 0.00001 | 0.00861 | +20.4% | 0.0485 | 0.08635 | 1.54 s | 3.42 s | 3.23 s |
| | | rand | 0.00823 | 0.00264 | 0.00001 | 0.00997 | +21.1% | 0.0793 | 0.1303 | 1.53 s | 2.17 s | 3.57 s |
| | | least | 0.00899 | 0.00304 | 0.00001 | 0.01096 | +21.9% | 0.1028 | 0.1680 | 1.74 s | 2.00 s | 3.87 s |
| | 2 | runner-up | 0.16338 | 0.05244 | 0.11015 | 0.19594 | +19.9% | 0.8689 | 1.2422 | 1.79 s | 2.58 s | 3.52 s |
| | | rand | 0.18891 | 0.06487 | 0.17734 | 0.22811 | +20.8% | 1.4231 | 1.8921 | 1.78 s | 1.96 s | 3.79 s |
| | | least | 0.20671 | 0.07440 | 0.23710 | 0.25119 | +21.5% | 1.8864 | 2.4451 | 1.98 s | 2.01 s | 4.01 s |
| | 1 | runner-up | 1.33794 | 0.58480 | 0.00114 | 1.58151 | +18.2% | 5.2685 | 10.079 | 1.87 s | 1.93 s | 3.34 s |
| | | rand | 1.57649 | 0.72800 | 0.00183 | 1.88217 | +19.4% | 8.9764 | 17.200 | 1.80 s | 2.04 s | 3.54 s |
| | | least | 1.73874 | 0.82800 | 0.00244 | 2.09157 | +20.3% | 11.867 | 23.910 | 1.94 s | 2.40 s | 3.72 s |
| CIFAR $5 \times [2048]$ | $\infty$ | runner-up | 0.00137 | 0.00020 | 0.00000 | 0.00167 | +21.9% | 0.0062 | 0.00950 | 18.2 s | 38.2 s | 33.1 s |
| | | rand | 0.00170 | 0.00030 | 0.00000 | 0.00212 | +24.7% | 0.0147 | 0.02351 | 19.6 s | 48.2 s | 36.7 s |
| | | least | 0.00188 | 0.00036 | 0.00000 | 0.00236 | +25.5% | 0.0208 | 0.03416 | 20.4 s | 50.5 s | 38.6 s |
| | 2 | runner-up | 0.06122 | 0.00948 | 0.00156 | 0.07466 | +22.0% | 0.2712 | 0.3778 | 24.2 s | 39.4 s | 41.0 s |
| | | rand | 0.07654 | 0.01417 | 0.00333 | 0.09527 | +24.5% | 0.6399 | 0.9497 | 26.0 s | 31.2 s | 42.5 s |
| | | least | 0.08456 | 0.01778 | 0.00489 | 0.10588 | +25.2% | 0.9169 | 1.4379 | 25.0 s | 33.2 s | 44.4 s |
| | 1 | runner-up | 0.93836 | 0.22632 | 0.00000 | 1.13799 | +21.3% | 4.0755 | 7.6529 | 24.7 s | 45.1 s | 40.5 s |
| | | rand | 1.18928 | 0.31984 | 0.00000 | 1.47393 | +23.9% | 9.7145 | 21.643 | 25.7 s | 36.2 s | 44.0 s |
| | | least | 1.31904 | 0.38887 | 0.00001 | 1.64452 | +24.7% | 12.793 | 34.497 | 26.0 s | 31.7 s | 44.9 s |
| CIFAR $6 \times [2048]$ | $\infty$ | runner-up | 0.00075 | 0.00005 | 0.00000 | 0.00094 | +25.3% | 0.0054 | 0.00770 | 27.6 s | 64.7 s | 47.3 s |
| | | rand | 0.00090 | 0.00007 | 0.00000 | 0.00114 | +26.7% | 0.0131 | 0.01866 | 28.1 s | 72.3 s | 49.3 s |
| | | least | 0.00095 | 0.00008 | 0.00000 | 0.00122 | +28.4% | 0.0199 | 0.02868 | 28.1 s | 76.3 s | 49.4 s |
| | 2 | runner-up | 0.03462 | 0.00228 | 0.00476 | 0.04314 | +24.6% | 0.2394 | 0.2979 | 37.0 s | 60.7 s | 65.8 s |
| | | rand | 0.04129 | 0.00331 | 0.01079 | 0.05245 | +27.0% | 0.5860 | 0.7635 | 40.0 s | 56.8 s | 71.5 s |
| | | least | 0.04387 | 0.00385 | 0.01574 | 0.05615 | +28.0% | 0.8756 | 1.2111 | 40.0 s | 56.3 s | 72.5 s |
| | 1 | runner-up | 0.59636 | 0.05647 | 0.00000 | 0.73727 | +23.6% | 3.3569 | 6.0112 | 37.2 s | 65.6 s | 66.8 s |
| | | rand | 0.72178 | 0.08212 | 0.00000 | 0.91201 | +26.4% | 8.2507 | 17.160 | 39.5 s | 53.5 s | 71.6 s |
| | | least | 0.77179 | 0.09397 | 0.00000 | 0.98331 | +27.4% | 12.603 | 28.958 | 40.7 s | 42.1 s | 72.5 s |
| CIFAR $7 \times [1024]$ | $\infty$ | runner-up | 0.00119 | 0.00006 | 0.00000 | 0.00148 | +24.4% | 0.0062 | 0.0102 | 8.98 s | 20.1 s | 16.2 s |
| | | rand | 0.00134 | 0.00008 | 0.00000 | 0.00169 | +26.1% | 0.0112 | 0.0218 | 8.98 s | 20.3 s | 16.7 s |
| | | least | 0.00141 | 0.00010 | 0.00000 | 0.00179 | +27.0% | 0.0148 | 0.0333 | 8.81 s | 22.1 s | 17.4 s |
| | 2 | runner-up | 0.05279 | 0.00308 | 0.00020 | 0.06569 | +24.4% | 0.2661 | 0.3943 | 12.7 s | 20.9 s | 20.7 s |
| | | rand | 0.05937 | 0.00407 | 0.00029 | 0.07496 | +26.3% | 0.5145 | 0.9730 | 12.6 s | 18.7 s | 21.8 s |
| | | least | 0.06249 | 0.00474 | 0.00038 | 0.07943 | +27.1% | 0.6253 | 1.3709 | 12.9 s | 20.7 s | 22.2 s |
| | 1 | runner-up | 0.76648 | 0.07028 | 0.00000 | 0.95204 | +24.2% | 4.815 | 7.9987 | 12.8 s | 21.0 s | 21.9 s |
| | | rand | 0.86468 | 0.09239 | 0.00000 | 1.09067 | +26.1% | 8.630 | 22.180 | 13.2 s | 19.8 s | 22.4 s |
| | | least | 0.91127 | 0.10639 | 0.00000 | 1.15687 | +27.0% | 11.44 | 31.529 | 13.3 s | 17.6 s | 22.9 s |

## E.2 Results on CROWN-general

Table 7: Comparison of certified lower bounds by CROWN-Ada on ReLU networks and CROWN-general on networks with tanh, sigmoid and arctan activations. CIFAR models with sigmoid activations achieve much worse accuracy than other networks and are thus excluded. For each norm, we consider the robustness against three targeted attack classes: the runner-up class (with the second largest probability), a random class and the least likely class.

| Network | | | Certified Bounds by CROWN-general | | | Average Computation Time (sec) | | |
|---|---|---|---|---|---|---|---|---|
| | $\ell_p$ norm | target | tanh | sigmoid | arctan | tanh | sigmoid | arctan |
| MNIST $3 \times [1024]$ | $\ell_\infty$ | runner-up | 0.0164 | 0.0225 | 0.0169 | 0.3374 | 0.3213 | 0.3148 |
| | | random | 0.0230 | 0.0325 | 0.0240 | 0.3185 | 0.3388 | 0.3128 |
| | | least | 0.0306 | 0.0424 | 0.0314 | 0.3129 | 0.3586 | 0.3156 |
| | $\ell_2$ | runner-up | 0.3546 | 0.4515 | 0.3616 | 0.3139 | 0.3110 | 0.3005 |
| | | random | 0.5023 | 0.6552 | 0.5178 | 0.3044 | 0.3183 | 0.2931 |
| | | least | 0.6696 | 0.8576 | 0.6769 | 0.3869 | 0.3495 | 0.2676 |
| | $\ell_1$ | runner-up | 2.4600 | 2.7953 | 2.4299 | 0.2940 | 0.3339 | 0.3053 |
| | | random | 3.5550 | 4.0854 | 3.5995 | 0.3277 | 0.3306 | 0.3109 |
| | | least | 4.8215 | 5.4528 | 4.7548 | 0.3201 | 0.3915 | 0.3254 |
| MNIST $4 \times [1024]$ | $\ell_\infty$ | runner-up | 0.0091 | 0.0162 | 0.0107 | 1.6794 | 1.7902 | 1.7099 |
| | | random | 0.0118 | 0.0212 | 0.0136 | 1.7783 | 1.7597 | 1.7667 |
| | | least | 0.0147 | 0.0243 | 0.0165 | 1.8908 | 1.8483 | 1.7930 |
| | $\ell_2$ | runner-up | 0.2086 | 0.3389 | 0.2348 | 1.6416 | 1.7606 | 1.8267 |
| | | random | 0.2729 | 0.4447 | 0.3034 | 1.7589 | 1.7518 | 1.6945 |
| | | least | 0.3399 | 0.5064 | 0.3690 | 1.8206 | 1.7929 | 1.8264 |
| | $\ell_1$ | runner-up | 1.8296 | 2.2397 | 1.7481 | 1.5506 | 1.6052 | 1.6704 |
| | | random | 2.4841 | 2.9424 | 2.3325 | 1.6149 | 1.7015 | 1.6847 |
| | | least | 3.1261 | 3.3486 | 2.8881 | 1.7762 | 1.7902 | 1.8345 |
| MNIST $5 \times [1024]$ | $\ell_\infty$ | runner-up | 0.0060 | 0.0150 | 0.0062 | 3.9916 | 4.4614 | 3.7635 |
| | | random | 0.0073 | 0.0202 | 0.0077 | 3.5068 | 4.4069 | 3.7387 |
| | | least | 0.0084 | 0.0230 | 0.0091 | 3.9076 | 4.6283 | 3.9730 |
| | $\ell_2$ | runner-up | 0.1369 | 0.3153 | 0.1426 | 4.1634 | 4.3311 | 4.1039 |
| | | random | 0.1660 | 0.4254 | 0.1774 | 4.1468 | 4.1797 | 4.0898 |
| | | least | 0.1909 | 0.4849 | 0.2096 | 4.5045 | 4.4773 | 4.5497 |
| | $\ell_1$ | runner-up | 1.1242 | 2.0616 | 1.2388 | 4.4911 | 3.9944 | 4.4436 |
| | | random | 1.3952 | 2.8082 | 1.5842 | 4.4543 | 4.0839 | 4.2609 |
| | | least | 1.6231 | 3.2201 | 1.9026 | 4.4674 | 4.5508 | 4.5154 |
| CIFAR-10 $5 \times [2048]$ | $\ell_\infty$ | runner-up | 0.0005 | - | 0.0006 | 37.3918 | - | 37.1383 |
| | | random | 0.0008 | - | 0.0009 | 38.0841 | - | 37.9199 |
| | | least | 0.0010 | - | 0.0011 | 39.1638 | - | 39.4041 |
| | $\ell_2$ | runner-up | 0.0219 | - | 0.0256 | 47.4896 | - | 48.3390 |
| | | random | 0.0368 | - | 0.0406 | 54.0104 | - | 52.7471 |
| | | least | 0.0460 | - | 0.0497 | 55.8924 | - | 56.3877 |
| | $\ell_1$ | runner-up | 0.3744 | - | 0.4491 | 46.4041 | - | 47.1640 |
| | | random | 0.6384 | - | 0.7264 | 54.2138 | - | 51.6295 |
| | | least | 0.8051 | - | 0.8955 | 56.2512 | - | 55.6069 |
| CIFAR-10 $6 \times [2048]$ | $\ell_\infty$ | runner-up | 0.0004 | - | 0.0003 | 59.5020 | - | 58.2473 |
| | | random | 0.0006 | - | 0.0006 | 59.7220 | - | 58.0388 |
| | | least | 0.0006 | - | 0.0007 | 60.8031 | - | 60.9790 |
| | $\ell_2$ | runner-up | 0.0177 | - | 0.0163 | 78.8801 | - | 72.1884 |
| | | random | 0.0254 | - | 0.0251 | 84.2228 | - | 83.1202 |
| | | least | 0.0294 | - | 0.0306 | 86.2997 | - | 86.9320 |
| | $\ell_1$ | runner-up | 0.3043 | - | 0.2925 | 78.7486 | - | 70.2496 |
| | | random | 0.4406 | - | 0.4620 | 89.7717 | - | 83.7972 |
| | | least | 0.5129 | - | 0.5665 | 87.2094 | - | 86.6502 |
| CIFAR-10 $7 \times [1024]$ | $\ell_\infty$ | runner-up | 0.0006 | - | 0.0005 | 20.8612 | - | 20.5169 |
| | | random | 0.0008 | - | 0.0007 | 21.4550 | - | 21.2134 |
| | | least | 0.0008 | - | 0.0008 | 21.3406 | - | 21.1804 |
| | $\ell_2$ | runner-up | 0.0260 | - | 0.0225 | 27.9442 | - | 27.0240 |
| | | random | 0.0344 | - | 0.0317 | 30.3782 | - | 29.8086 |
| | | least | 0.0376 | - | 0.0371 | 30.7492 | - | 30.7321 |
| | $\ell_1$ | runner-up | 0.3826 | - | 0.3648 | 28.1898 | - | 27.1238 |
| | | random | 0.5087 | - | 0.5244 | 29.6373 | - | 30.5106 |
| | | least | 0.5595 | - | 0.6171 | 31.3457 | - | 30.6481 |