[Reviews · NeurIPS 2018]

Reviewer 1



- The problem is interesting and worth studying. - The author(s) propose a solution for general activation functions. This problem is still unexplored before.

Reviewer 2



Summary: This paper proposes a general framework CROWN to efficiently certify robustness of neural networks with general activation functions. CROWN adaptively bounds a given activation function with linear and quadratic functions, so it can tackle general activation functions including but not limited to the four popular choices: ReLU, tanh, sigmoid, and arctan. Experimental results demonstrate the effectiveness, efficiency, and flexibility of the proposed framework. Quality: We are glad to find a work which conducts the efficiently certifying of the non-trivial robustness for general activation functions in neural networks. It is also interesting that the proposed framework can flexibly select upper bounds and lower bounds which can reduce the approximation error. This paper is of high quality. The problem is well-motivated, and the experimental comparisons and analysis are sufficient and complete. The proofs of the theorems and bounds are reasonable. Clarity: This paper is very well written and enjoyable to read. The theory analysis and proof are clear and easy to follow. Originality: This is an original research work especially of the robustness certification with the general activation functions, which may provide a good direction for the future research on this problem. Overall impression, this is a high-quality paper with solid analysis, proof, and experimental results.

Reviewer 3



The paper proposes a framework (CROWN) to efficiently compute for the minimum distortion for a given neural network using linear (and quadratic) upper bounds for the activation functions (that might not be piece-wise linear), generalizing some earlier approaches such as Fast-Lin. The experimental results demonstrate quality comparable with the earlier algorithms. The paper is well-written, interesting and about a very relevant topic, but the results might be to incremental for NIPS. Minor comments: 97: It would be helpful to elaborate/highlight further what fast means (such as computational complexity). 97: What does "certified" mean in this context? Does the process result in some verifiable certificate?